# Climatic and tectonic drivers shaped the tropical distribution of coral reefs

Lewis A. Jones [1✉], Philip D. Mannion [2], Alexander Farnsworth [3], Fran Bragg[3] & Daniel J. Lunt [3]

Today, warm-water coral reefs are limited to tropical-to-subtropical latitudes. These diverse ecosystems extended further poleward in the geological past, but the mechanisms driving these past distributions remain uncertain. Here, we test the role of climate and palaeogeography in shaping the distribution of coral reefs over geological timescales. To do so, we combine habitat suitability modelling, Earth System modelling and the ~247-million-year geological record of scleractinian coral reefs. A broader latitudinal distribution of climatically suitable habitat persisted throughout much of the Mesozoic–early Paleogene due to an expanded tropical belt and more equable distribution of shallow marine substrate. The earliest Cretaceous might be an exception, with reduced shallow marine substrate during a 'cold-snap' interval. Climatically suitable habitat area became increasingly skewed towards the tropics from the late Paleogene, likely steepening the latitudinal biodiversity gradient of reef-associated taxa. This was driven by global cooling and increases in tropical shallow marine substrate resulting from the tectonic evolution of the Indo-Australian Archipelago. Although our results suggest global warming might permit long-term poleward range expansions, coral reef ecosystems are unlikely to keep pace with the rapid rate of anthropogenic climate change.

[1] Centro de Investigación Mariña, Grupo de Ecoloxía Animal, Universidade de Vigo, 36310 Vigo, Spain. [2] Department of Earth Sciences, University College London, Gower Street, London WC1E 6BT, UK. [3] School of Geographical Sciences, University of Bristol, Bristol BS8 1SS, UK. ✉email: LewisAlan.Jones@uvigo.es

Despite covering <0.1% of the oceans, warm-water coral reefs support the greatest biodiversity of marine organisms on Earth[1–3]. These rich ecosystems are limited to tropical and subtropical latitudes (~34°N–32°S), with minimum sea surface temperature (~18 °C) tolerances being the primary constraint on this distribution[4–7]. Accordingly, many reef-associated organisms (e.g. reef corals and reef fishes) are also largely restricted to low latitudes, with their diversity concentrated in the tropics and subtropics, contributing to a steep latitudinal biodiversity gradient[8–11]. A substantial proportion of this biodiversity is found in the Indo-Australian Archipelago (i.e. ~75% of zooxanthellate corals[12]), with the region recognised as a marine biodiversity hotspot[3,13,14].

In contrast to their present-day distribution, fossil data suggest that coral reefs were less skewed towards the tropics in the geological past, with reefs found at higher latitudes during some time intervals[15–17]. For example, a fossil coral reef of early–middle Eocene (56–41.2 millions of years ago [Ma]) age has been discovered at a palaeolatitude of 46°N[15]. Both abiotic (e.g. climate) and biotic (e.g. competition and biological adaptation) factors have been proposed as potential drivers of these past range shifts[17–19]. Due to the limited sea surface temperature tolerances of warm-water coral reefs today[5], past latitudinal shifts are most often considered to reflect an associated latitudinal expansion of tropical and subtropical conditions[15,17,20]. Nevertheless, plate tectonics are also thought to be a fundamental driver of biodiversity over geological timescales, modulating the distribution of the continents and shallow marine substrate[14,21–25], and thus the distribution of coral reefs and their associated biota too:[14,18,21,22] if there is insufficient suitable benthic substrate for shallow marine taxa, the climatic conditions might be irrelevant for reef development (e.g. ref. [26]).

Given that both climate and the distribution of shallow marine area have substantially varied throughout geological time, the question arises as to the degree to which these factors can explain the observed distribution of fossil coral reefs. Moreover, understanding how these drivers influenced the spatiotemporal distribution of climatically suitable habitat area for coral reefs might explain their tropically-skewed distribution today. The strong relationship between the biodiversity of reef-associated organisms and reef habitat area observed in the present[8,27–30], as well as the capacity of ancient reefs to serve as major sources ('cradles') of biodiversity[31], suggests past latitudinal shifts in available reef habitat might have shaped the distribution of reef-associated biodiversity in deep time. Specifically, the loss of high latitude suitable habitats in the past might explain the steepness of the latitudinal biodiversity gradient in reef-associated biota today. Coral reefs are currently rapidly deteriorating as a result of rising sea surface temperatures and associated bleaching[32–37]. Over the next century, increasing atmospheric $CO_2$ levels are expected to further exacerbate the environmental pressures on coral reef ecosystems through ocean acidification[38,39]. With projected global warming of 1.8–5.6 °C and atmospheric $CO_2$ levels up to 1100 ppm by 2100 AD[40], severe repercussions are predicted for coral reefs, and the rich biodiversity they house[33,36,38,39]. Thus, understanding their response to climatic change has never been more imperative.

The fossil record provides a unique opportunity to evaluate—with empirical evidence—the long-term response of coral reefs to past climatic shifts. Sea level and temperature fluctuations shaped the distribution of coral reef ecosystems during the Quaternary, i.e. the last ~2.6 million years (myr)[41–43]. However, previous studies have failed to find strong correlations between the distribution of reefs and inferred abiotic controls prior to the Quaternary[15,17,18]. Considering the general patchiness of the fossil record, as well as preferential sampling of the Northern Hemisphere[18,44–46], this is perhaps not unexpected. In fact, one study found that the single most important factor explaining the sampled distribution of ancient reefs was gross domestic product, with the majority of fossil reef data stemming from wealthy countries[18]. Moreover, recent work has shown that palaeo-temperature reconstructions–based on proxy data–are also influenced by similar sampling biases, which might have impacted the interpretations of previous studies investigating the influence of climatic drivers on climatically sensitive ecosystem[47]. Unfortunately, such biases substantially limit our understanding of the drivers of macroevolutionary patterns in ancient reefs. Determining the full extent of their past global distribution, based solely on their fossil record, is a near-impossible challenge without greater spatiotemporal coverage.

Here, using habitat suitability modelling, we test whether the present-day climatic tolerances of warm-water coral reefs can explain their geographic distribution in the geological past. To do so, we integrate present-day coral reef occurrence data with Earth system modelling to predict the distribution of climatically suitable habitat area since the earliest proliferation of Scleractinia (stony corals) ~247 Ma[19,48,49]. Using our model outputs, we test the capacity of our habitat suitability model (HSM) to predict the presence of known fossil coral reef localities. Building upon this, we infer how the latitudinal distribution of warm-water coral reefs and associated taxa might have evolved during the last ~247 myr. Our novel approach provides insight into the role played by climate and palaeogeography in shaping the distribution of warm-water coral reefs and their associated biodiversity.

## Results

**Model performance.** Using Maxent[50,51], we modelled the distribution of climatically suitable habitat, within shallow marine environments (<200 m depth), for warm-water coral reefs from the Anisian (Middle Triassic) to the Piacenzian (latest Neogene), i.e. the last ~247 myr. We calibrated our HSM using the present-day distribution of warm-water coral reefs (Fig. 1) and climatic layers (see Methods). Subsequently, we projected the model onto stage-level estimates of past abiotic conditions and evaluated the capacity of model hindcasts to correctly predict the distribution of fossil coral reef localities (n = 535) for each stage (e.g. intersecting Piacenzian fossil coral reef localities with the Piacenzian hindcast).

Our modern-trained HSM scored an average of 0.856 (standard deviation: 0.003) for the area under the curve (AUC) statistic[52] and 0.988 (standard deviation: 0.006) for the continuous Boyce index[53,54], suggesting a good discriminatory capacity for the modern. Stage-level binary hindcasts demonstrate a moderate to high predictive performance, with an average of ~60–87% (standard deviation: ~19–29%) of fossil reef localities accurately predicted by model hindcasts, depending on the stage and binary threshold selection (Fig. 2; Table S2–S3; Figs. S2–3). When considering a single cell search buffer around fossil reef localities, these values increase to ~62–90% (standard deviation: ~17–28%) (Fig. 2; Table S2–S3; Figs. S2–3). This predictive performance was not predicated by geographically vast estimates of suitable habitat area across the Earth. Based on randomly generated datasets of spatial points, on average, only ~3–8% of points would randomly intersect with model hindcasts across all stages, compared to the ~60–87% for fossil reef localities (Fig. 2; Table S2–3; Figs. S2–3). This observation is supported by one-sample (one-sided) Wilcoxon rank-sum tests (see Table S2–S3), which suggest that the percentage of randomly generated points intersecting with suitable habitats is significantly less than those of fossil reef localities for every stage (P < 0.001), except for the Bartonian (middle Eocene; P = 1) under both binary thresholds,

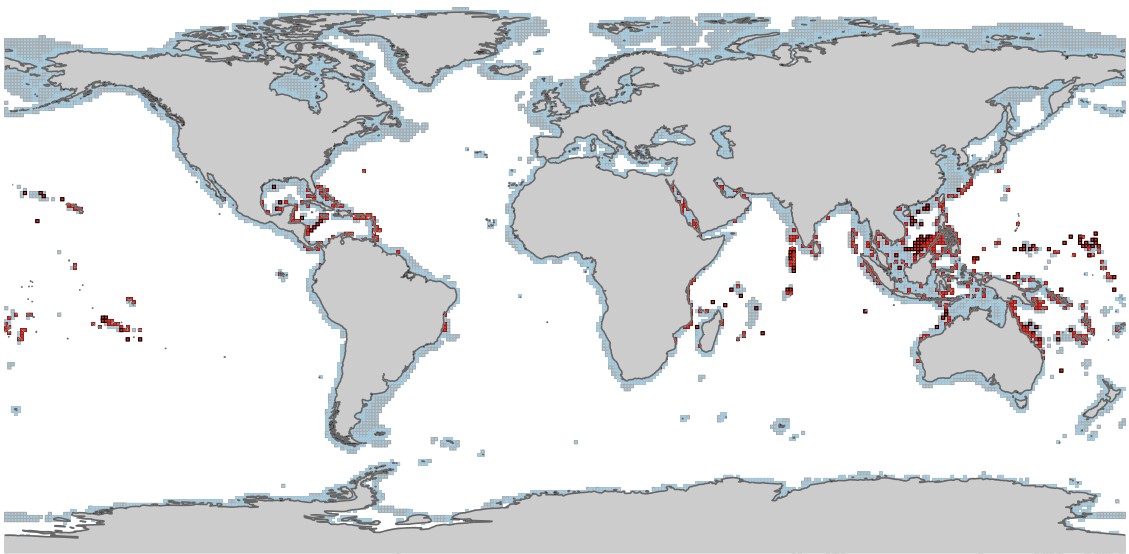

**Fig. 1 Global map of present-day warm-water coral reefs from the ReefBase (http://www.reefbase.org/main.aspx).** Distribution data are spatially subsampled at a horizontal resolution of 1° × 1° and filtered to include only 'true reefs' (see Methods). Presence cells ($n = 790$) are depicted in red (cells containing at least one coral reef or community). Shallow-water mask cells ($n = 11,221$) are depicted in blue, and denote areas of substrate depth <200 m. These cells were randomly sampled to generate background data and calibrate the habitat suitability model. Continents are depicted in light grey.

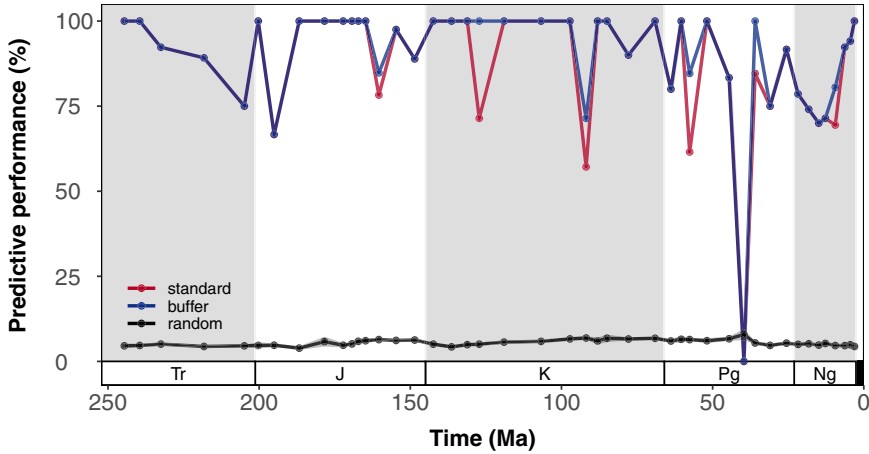

**Fig. 2 Predictive performance of stage-level (Anisian, Triassic to Piacenzian, Neogene) hindcasts from habitat suitability modelling for binary threshold 'LTP' (least training presence).** Predictive success of fossil reef localities (percentage of total fossil coral reef localities intersecting with cells predicted to be suitable) is indicated by the red points; predictive success of fossil reef localities with a one-cell buffer (one-cell queen moves) is indicated by blue points; predictive success from random spatial point generation is indicated by the black points, along with the 95% confidence intervals. Period abbreviations are as follows: Triassic (Tr), Jurassic (J), Cretaceous (K), Paleogene (Pg), and Neogene (Ng).

as well as the Hettangian, Sinemurian, Bathonian (Early–Middle Jurassic) and Selandian (Paleocene) under MaxSSS ($P > 0.05$). Furthermore, we found that for 82% (37/45) of evaluated stages, continuous suitability values associated with known fossil reef localities were significantly greater than those expected under random distributions within our shallow-water masks (two-sample (one-sided) Wilcoxon rank-sum tests; $P < 0.05$; Fig. S4).

Whilst the predictive performance of model hindcasts was high on average, there are notable temporal differences in performance, with a standard deviation of ~19–29% across all stages (Fig. 2; Table S2–S3; Fig. S2–S3). Notably, more recent stages (last 37 myr [late Eocene onwards]) generally have a poorer predictive performance under binary threshold MaxSSS than the rest of the time series, though generally still significantly better than random (one-sample (one-sided) Wilcoxon rank-sum tests: $P < 0.05$).

Overall, most false negatives (71 out of 73) in model hindcasts are due to fossil reef localities intersecting with low sea surface temperature values (<18 °C) in the mean minimum sea surface temperature layer. Multivariate Environmental Similarity Surfaces analyses[55] indicate that 80 fossil reef localities are found in novel environmental conditions outside the calibration range of the modern-trained HSM. Notably, 62 of these localities intersects with mean maximum sea surface temperatures above those of the present-day climatic landscape (33.8–35.6 °C).

**Habitat suitability model predictions**. The centroid of Northern Hemisphere binary predictions shows a long-term equatorward shift in the distribution of suitable habitats from 24–29° in the Sinemurian (Early Jurassic) to 13–17° in the Piacenzian (Fig. 3; Fig. S5). By contrast, the Southern Hemisphere centroid expresses

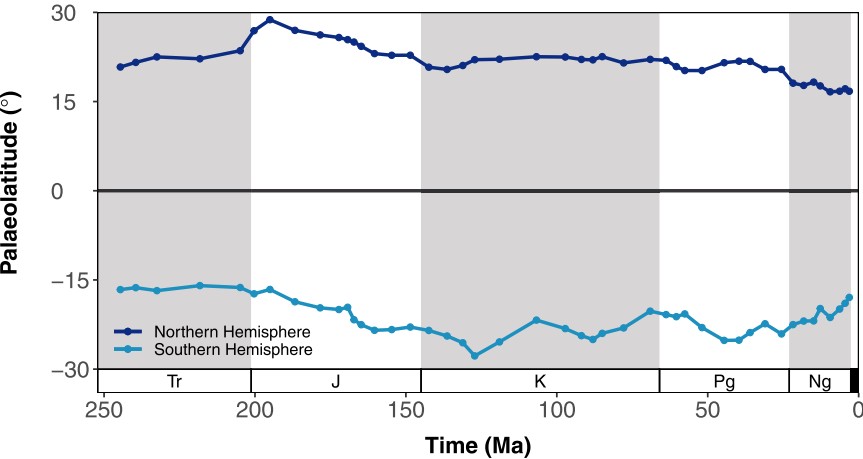

**Fig. 3 Palaeolatitudinal shifts in the centroid of suitable habitats for warm-water coral reefs from the Anisian (Triassic) to the Piacenzian (Neogene).** The centroid was computed from binary suitability maps under threshold 'LTP' (least training presence). Centroid calculation was carried out for each hemisphere and implemented with weights proportional to the area of each suitable cell to account for variable cell area with latitude. Period abbreviations are as follows: Triassic (Tr), Jurassic (J), Cretaceous (K), Paleogene (Pg), and Neogene (Ng).

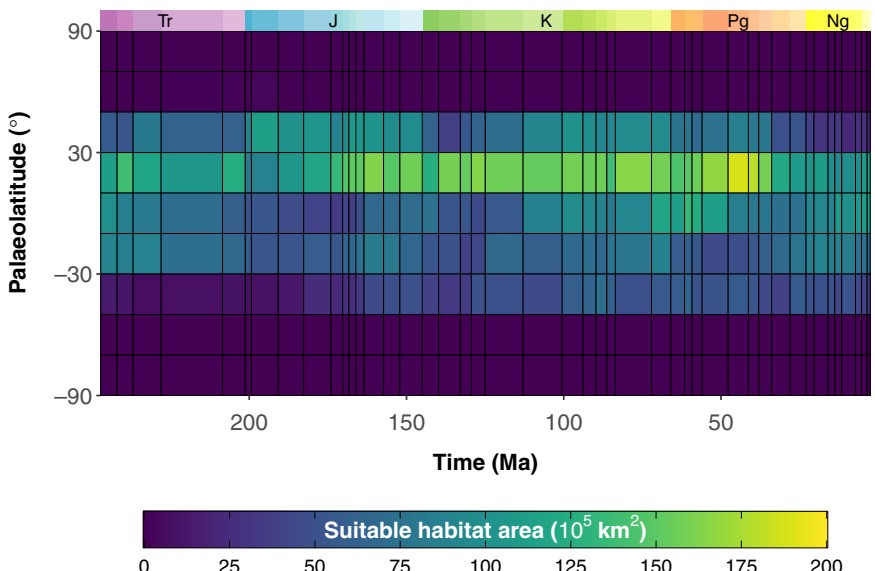

**Fig. 4 Palaeolatitudinal (20° bins) estimates of climatically suitable habitat area (Anisian, Triassic to Piacenzian, Neogene) for warm-water coral reefs under binary threshold 'LTP' (least training presence).** Stage-level hindcasts are based on a modern-calibrated Maxent model and area is calculated from binary predictions. Period abbreviations are as follows: Triassic (Tr), Jurassic (J), Cretaceous (K), Paleogene (Pg), and Neogene (Ng).

greater volatility, with prominent equatorward and poleward palaeolatitudinal shifts in the centroid of suitable habitat area (Fig. 3; Fig. S5). However, both hemispheres demonstrate a prominent equatorward shift (~5–6°) in climatically suitable habitat area since the Priabonian (late Eocene) onwards. The Northern Hemisphere shift is driven by a ~54–72% decline in climatically suitable habitat area at 30–50°N, resulting in suitable habitat area becoming increasingly skewed towards the northern tropics/subtropics (Fig. 4; Fig. S6). This is driven by a decrease in sea surface temperature at temperate latitudes (Fig. S7). By contrast, the Southern Hemisphere equatorward shift is driven by an increase in suitable habitat area in the southern tropics/subtropics (Fig. 4; Fig. S6). This increase is in response to an expansion of available shallow marine substrate area at low latitudes in the shallow-water masks (Fig. S8). Palaeolatitudinal analyses suggest that the areal extent of climatically suitable habitat was variable throughout most of the last ~247 myr, though the majority of the suitable habitat area occurred between 10 and 30°N. Nevertheless, for most of the Mesozoic and early Cenozoic, high latitude

(30–50°N) climatically suitable habitat was more readily available than in the present day (Fig. 4; Fig. S6). The earliest Cretaceous (Berriasian–Valanginian) is one notable exception, with a 37% reduction in shallow marine area across the Jurassic/Cretaceous (J/K) boundary (Fig. S8). Global analyses also suggest similar temporal variation in the areal extent of suitable habitat area (Fig. 5). In general, high global habitat area is modelled for the Late Jurassic and Late Cretaceous–Eocene. However, major declines in habitat area are estimated across the J/K boundary, as well as from the Priabonian onwards. This is largely due to the loss of high latitude (30–50°N) climatically suitable habitat area (Fig. 4). Finally, ordinary least-squares regression analyses suggest that there is no significant relationship between the number of fossil coral reef sites and the availability of climatically suitable habitat ($R^2 = 0.000–0.001$, $P = 0.878–979$; Fig. S11).

Stage-level estimates of the distribution of suitable habitat area suggest that the reef zone has varied substantially over the last ~247 myr (Fig. 6; Fig. S9). The modelled reef zone reached a maximum palaeolatitudinal extent in the Bartonian, ranging from

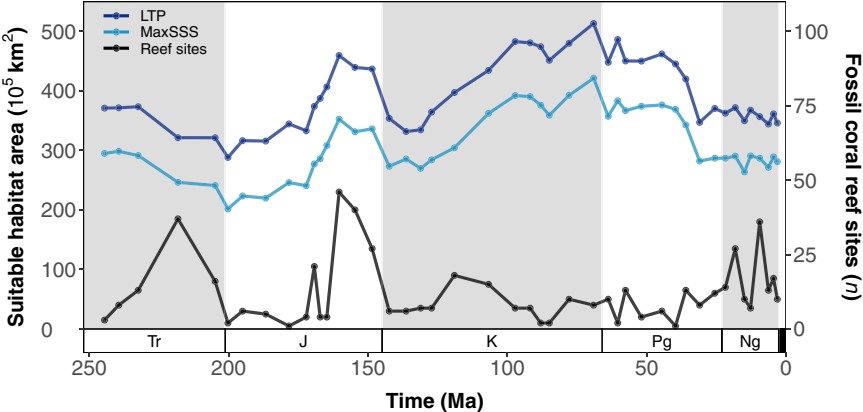

**Fig. 5 Global estimates of suitable habitat area (Anisian to Piacenzian) for warm-water coral reefs under binary thresholds 'LTP' (least training presence; dark blue) and 'MaxSSS' (maximising the sum of sensitivity and specificity; light blue).** Stage-level hindcasts are based on a modern-calibrated MaxEnt model, and area is calculated from binary predictions (LTP/MaxSSS). The number of fossil warm-water coral reef sites is depicted in black. Ordinary least-squares regression analyses suggests that there is no significant relationship between the number of fossil coral reef sites and the availability of climatically suitable habitat (see Fig. S11). Period abbreviations are as follows: Triassic (Tr), Jurassic (J), Cretaceous (K), Paleogene (Pg), and Neogene (Ng).

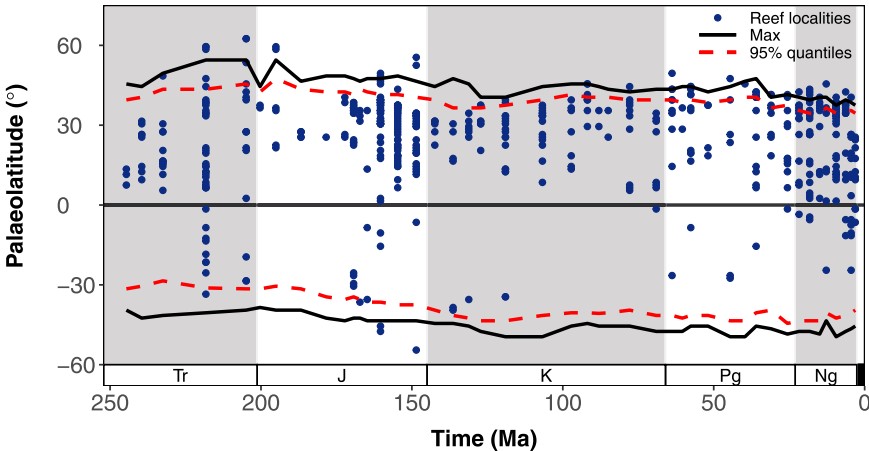

**Fig. 6 Stage-level estimates of warm-water coral reef zone from the Anisian (Triassic) to the Piacenzian (Neogene).** Reef zone (black lines) is defined as the most poleward palaeolatitude of suitable habitat in each hemisphere (North and South) under binary threshold 'LTP' (least training presence). The 95% quantiles of estimated stage-level reef zone are depicted by the dashed red line, and fossil coral reef localities are indicated by the blue points. Period abbreviations are as follows: Triassic (Tr), Jurassic (J), Cretaceous (K), Paleogene (Pg), and Neogene (Ng).

46.5°N to 49.5°S, under the LTP binary threshold (Fig. 6). However, under binary threshold MaxSSS, the Albian (mid-Cretaceous) had the largest reef zone, ranging from 36.5°N to 43.5°S (Fig. S9). Nevertheless, under both thresholds, the reef zone is estimated to have a minimum palaeolatitudinal extent during the Piacenzian, ranging from 29.5–37.5°N to 36.5–45.5°S (Fig. 6; Fig. S9). The 95% quantiles of estimated reef zone indicate largely confirmative temporal patterns in the palaeolatitudinal extent of the reef zone, except for the earliest Cretaceous in the Northern Hemisphere which indicates a latitudinally diminished reef zone (Fig. 6; Fig. S9). Overall, the total palaeolatitudinal extent of the modelled reef zone declined by ~11–12° from the Norian (Late Triassic) towards the Piacenzian. Although most fossil coral reef localities are found within the estimated reef zone for our entire time series (~73–92%), several high palaeolatitude reef localities are found outside the estimated reef zone, particularly in the Northern Hemisphere (Fig. 6; Fig. S9).

## Discussion

Our results suggest that coral reefs largely tracked major latitudinal shifts in tropical and subtropical conditions throughout the Mesozoic and Cenozoic. For much of the last ~247 myr, suitable habitat area was notably less skewed towards the tropics than today. However, the earliest Cretaceous (Berriasian–Valanginian) might be one exception. Across the J/K boundary, suitable habitat area declined substantially at 30–50°N in response to a 37% decrease in available shallow marine substrate (Fig. S8). This decline at temperate latitudes resulted in Northern Hemisphere suitable habitat area being skewed towards the tropics during the earliest Cretaceous (Fig. 4; Fig. S6), though this was less extreme than in the present day. Interestingly, this interval is characterised by a major fall in global sea level[56], reducing the extent of continental flooding, and hence available shallow marine substrate. Although not explicitly modelled in our climate simulations, the earliest Cretaceous has also been interpreted to have experienced geologically brief 'cold snaps' (e.g. refs. [57–61]), and has occasionally been referred to as a 'cool greenhouse' period[62]. Previous work has suggested that the observed J/K sea level fall might be the result of polar ice development associated with the onset of these cooling episodes (e.g. ref. [56]). If true, the earliest Cretaceous cold snaps might have had both a direct and indirect effect on the distribution of coral reefs. Regardless, the high latitude (30–50°N) reduction in suitable habitat might have led to heightened turnover in marine ecosystems (e.g. ref. [63]).

Palaeolatitudinal analyses suggest that climatically suitable habitat area was predominately concentrated in the Northern Hemisphere throughout most of the Mesozoic and early Cenozoic, particularly between 10 and 30ºN. This is noteworthy as previous work has frequently noted that there is a major global sampling bias towards the Northern Hemisphere, especially in western Europe and North America (e.g. refs. [18,44–46]). Although we do not challenge this view, this skew of suitable habitat area suggests that greater sampling opportunities might be available in the Northern Hemisphere for fossil coral reefs. In combination with preferential sampling, this skew of suitable habitat area towards the Northern Hemisphere might further explain the relative scarcity of fossil reefs reported from the Southern Hemisphere (Fig. 6). Such biases might also explain why an insignificant and weak relationship is found between climatically suitable habitat area and the number of reef sites in this study. However, this might also be explained by other factors such as biotic interactions (e.g. competition displacement) or evolutionary crises (e.g. mass extinctions).

Warm-water coral reef habitat became increasingly skewed towards the tropics from the late Paleogene (~37 Ma) to the present day (Figs. 3–4; Fig. 6; Figs. S4–5). This is partly due to a decline in suitable habitat area at temperate latitudes (30–50ºN), resulting from the onset of cooler sea surface temperatures there (Fig. S7) (e.g. refs. [59,64,65]). This broadly supports previous work in suggesting that the modern latitudinal biodiversity gradient likely steepened during the late Paleogene as a consequence of the onset of 'icehouse' climatic conditions[21,66–69]. However, our results also lend support for a fundamental role in the tropical concentration of biodiversity played by changes in continental configuration. Suitable habitat area in southern tropical/subtropical latitudes increased from the late Paleogene to the present day due to the equatorward shift of the Australian Plate, and its eventual convergence with the Eurasian and Philippine plates[21,70,71]. This reconfiguration led to an increase in shallow marine substrate area in the tropics (Fig. S8). In this regard, our results also support previous work in suggesting that plate tectonics enabled the formation of the Indo-Australian Archipelago biodiversity hotspot[14,21,22], and thus contributed to the tropical concentration of reef-associated biota[22,72]. This finding is notable as the distribution of shallow marine substrate has substantially changed throughout geological time, in response to changes in continental configuration. During supercontinent phases (e.g. Pangaea), the loss of coastlines led to a reduction in shelf area that limited the availability of suitable habitat area (e.g. ref. [23,73]). This might have been particularly limiting for climatically-sensitive organisms, such as zooxanthellate corals, especially when the supercontinent was aligned in a North–South orientation (e.g. during the early Mesozoic). Ultimately, this might have shaped broad-scale temporal and spatial biodiversity patterns over geological timescales (e.g. ref. [73]). Although our models do not inform directly about the distribution of biodiversity, our results support the view that the present-day unimodal latitudinal biodiversity gradient became increasingly steep from the late Paleogene onwards. The nature of latitudinal biodiversity gradients prior to this is more difficult to decipher[46]. However, flattened gradients might have persisted throughout the majority of the Mesozoic (perhaps excluding the earliest Cretaceous) and early Paleogene due to an expanded reef zone (Figs. 3–4; Fig. 6), at least for reef-associated taxa.

While the vast majority of fossil coral reefs were accurately predicted by our modern-trained HSM, 71/535 fossil coral reef localities intersected with minimum sea surface temperatures below their present-day tolerance (18 °C). This might be a result of a mid- and high-latitude cold bias present in the paleoclimatic simulations (e.g. ref. [74]), resulting in temperate sea surface

temperatures being underestimated for some intervals, such as the last 37 myr. However, our results also indicate that 62/535 localities intersected with maximum sea surface temperatures (33.8–35.6 °C) above the upper thermal limit of the oceans today. Together, these findings suggest that coral reefs might have occupied a broader climatic space during intervals of the geological past. Although this might offer some optimism for the current fate of coral reefs in a warming world, caution is required. These coral reef ecosystems developed in a greenhouse world, with ecological communities adapted to warmer climatic conditions which evolved over millions of years. Although it is possible that coral reefs might fare better under projected global warming than current model estimates (e.g. refs. [38,75]), the inherent biological richness and economic advantages provided by coral reefs demands conservatism in such estimates.

Our study is not without limitations. Firstly, our climate simulations use highly idealised $CO_2$ concentrations, in particular for the pre-Cenozoic, where $CO_2$ is held constant. Although these concentrations capture temporal differences between the broadest-scale climate states (i.e. 'icehouse' vs. 'greenhouse' intervals), stage-on-stage differences (and within stage) in $CO_2$ levels exist (e.g. ref. [76]), which are not captured here. As such, climate variability within stages is not captured within our palaeoclimatic simulations. However, our simulations do incorporate varying palaeography, which is important for local temperature and hydrological cycle changes[77]. Secondly, our results might be sensitive to the climate model of choice. Although it has been shown that climate models generate robust estimates of past climate at the broadest scale, inter-model geographic differences exist, which might influence the results of this study[74,78,79]. It might be preferable to use an ensemble of palaeoclimate data from various models in future work. However, only data from HadCM3L are available for our entire study period. Despite this, differences in palaeoclimatic reconstructions between climate models are unlikely to be so great that they affect the overall conclusions presented here. Thirdly, the spatial resolution (1° × 1°) of this study might also bias estimates of suitable habitat area (e.g. ref. [80]). For example, bathymetry could vary considerably within 1° × 1° cells (e.g. on a carbonate ramp or atoll), with differing proportions of shallow marine substrate between cells. This issue is not limited to deep time studies with global neontological studies frequently using similar spatial resolutions (e.g. refs. [75,81]). Furthermore, the native horizontal resolution of the climate model data (2.5° × 3.75°) may fail to resolve some local-scale variation in climatic variables, such as sea surface temperature. However, the use of finer spatial resolutions in deep time is dependent on the ability to robustly estimate abiotic conditions at such resolutions. This is often challenging due to the limited spatiotemporal empirical evidence provided by the geological record[47]. Although fossil reef records ($n = 535$) suggest that our model hindcasts performed generally well across the entire time series, it is notable that several stages had very few records to evaluate specific stage-level hindcasts. This limitation could be lessened in the future with improved sampling and age estimates of fossil reefs. Finally, our analyses also assume that all climatically suitable habitat area was accessible and occupied, which is not necessarily the case due to dispersal limitations (e.g. restricted seaways, ocean circulation patterns, and the distribution of the continents) and variables not included in our models (e.g. upwelling, nutrient concentrations, aragonite saturation and siliciclastic shedding zones). The omission of some ecologically-important variables in our study (e.g. nutrient concentrations) was principally dictated by the lack of data availability, and likely results in our suitability models being less constrained than otherwise. This is best exemplified by suitable habitat being estimated along the west coast of Africa: although coral

communities are known to exist in this region, local environmental conditions (e.g. upwelling and siliclastic shedding) prevent the formation of 'true reefs' here[82,83]. Whilst thermal upwelling is explicitly resolved in the palaeoclimate model used in this study, elevated nutrient concentrations associated with upwelling zones are not. The work presented here focuses on whether coral reefs tracked climatic changes over geological timescales. In future work, the inclusion of such estimates might help to further constrain coral reef distributions in deep time when reasonable approximations are available. However, previous work has also demonstrated that some variables might be less relevant at larger spatial resolutions (such as ours), and reducing the number of variables during model calibration is vital to prevent model overfitting (e.g. ref. [81]). Nevertheless, our estimations of past suitable habitat area offer the potential to disentangle whether observed coral reef distributions in the fossil record were the result of climatic drivers, or some alternative mechanism.

Despite these potential limitations, we are able to predict the distribution of fossil reef localities with high fidelity (~60–87%) using habitat suitability modelling. Considering this high predictive performance, this approach could be utilised to identify suitable areas to target for fossil prospecting (e.g. ref. [84]). The predictive performance of our HSM also suggests that climate and palaeogeography both played a pivotal role in shaping the distribution of coral reefs over the last ~247 million years. Given the current rapid rate of climate change, major shifts in the distribution of suitable habitat for reef development should be expected, as has been suggested in previous work[38,75,85]. However, while coral reef ecosystems might have been able to track geographic shifts in suitable conditions over geological time intervals, they are unlikely to keep pace with the rapid rate of anthropogenically-driven climate change[38,75].

## Methods

**Coral reef occurrence data set**. The global distribution of present-day warm-water coral reefs was downloaded from 'ReefBase' (http://www.reefbase.org/main.aspx), a composite database of published and unpublished sources of coral reef localities. This database includes over 10,000 point coordinates and has been frequently utilised in coral reef studies, e.g. refs. [38,81,86]. For each database entry, supplementary data are available on reef type, such as 'fringing reef' or 'non-reef community'. For the purpose of this study, we restricted our analyses to 'true reefs', and removed all 'non-reef coral community' entries as these are characterised by the inability to accumulate calcium carbonate. Further quality control was performed on this dataset, whereby occurrences found to occur on land or at depths >200 m (approximating the photic zone) were removed. Prior to analysis, point occurrences were spatially subsampled to a grid resolution of 1° × 1° to correspond with climatic layers, and prevent multiple records being excessively weighted during model training. The final dataset consists of 790 coral reef localities for model calibration (Fig. 1).

**Climate data**. To estimate the climatic tolerances of present-day warm-water coral reefs, two climatic variables (sea surface temperature and sea surface insolation) were considered. These variables are known to be limiting to the latitudinal distribution of warm-water coral reefs today[5,81,87], and can be viably determined for the geological past using palaeoclimatic modelling. Initially, these variables were considered at various temporal scales (e.g. monthly, seasonal, annual), as well as summary derivatives (e.g. range, maximum, minimum), calculated on a cell-by-cell basis from the climate model results. However, to reduce collinearity between variables and prevent over-fitting[88], we retained a combination of only four climatic variables: mean maximum sea surface temperature, mean minimum sea surface temperature, mean maximum insolation, and mean minimum insolation.

Present-day (pre-industrial) and stage-level climate simulations for the Anisian (Middle Triassic) to the Piacenzian (latest Neogene) were carried out using the HadCM3BL-M2.1aE model, a version of the HadCM3L coupled atmosphere-ocean general circulation model[89]. These model components are capable of resolving key features such as the Hadley-Walker circulation, wind driven and thermohaline circulatory systems, as well as gyres and upwelling, which will have an impact on coral reef distributions. The HadCM3L climate model has a horizontal resolution of 2.5° latitude × 3.75° longitude in the atmosphere and ocean, with a vertical resolution of 19 levels in the atmospheric component and 20 levels (5550 m depth) in the oceanic component[89]. It is similar to the HadCM3 climate model, which has been utilised in numerous ecological studies (e.g. refs. [85,90,91]), but differs in its

reduced ocean resolution. This divergence from the HadCM3 is necessary due to the substantial spin-up times required for climate simulations to equilibrate to the radically different climates and boundary conditions (land-sea distribution, topography, bathymetry, solar luminosity, land-ice distribution and $pCO_2$) in deep time[92]. Despite this, HadCM3L has been shown to perform well in reproducing average global and regional scale climate patterns recorded in proxies[77,89,93]. Recently, HadCM3L has been used in a number of palaeobiological studies[73,94–97], and has demonstrated a capacity to predict the distribution of climatically-sensitive organisms[69,94]. For our stage-level climate simulations, the $CO_2$ concentration was held constant from the Triassic to Eocene at 1120 ppmv, decreasing to 560 ppmv during the Oligocene, 400 ppmv during the Miocene and early Pliocene, and finally 280 ppmv during the late Pliocene and pre-industrial (Table S1). These values are within the range of uncertainty of a multi-proxy compilation of atmospheric $CO_2$ (i.e. ref. [76]), but should be considered as idealised. Each stage-level palaeoclimatic simulation was run for 1422 years, reaching near-surface equilibrium[77], and were run in an identical way to ref. [98]. For the purposes of this study, climate layers were downscaled to a horizontal resolution of 1° × 1° using bilinear interpolation.

**Shallow-water mask**. All climatic variables were clipped by shallow-water masks (<200 m substrate depth, approximating the photic zone) prior to modelling. This was done to prevent artificially inflating model validation scores by training on large spatial extents[99]. To generate our shallow-water masks, we extracted bathymetric data from Getech's (https://getech.com) digital elevation models (DEMs), which provide global gridded (0.5° × 0.5°) representations of the Earth's topography and bathymetry at stratigraphic stage level (Fig. S12). These DEMs have been used in a number of deep-time applications (e.g. refs. [73,100,101]), and are utilised as the boundary conditions for climate simulations, providing spatially explicit data for when continent configuration, topography, and bathymetry were different from today[102]. To correspond with climate layers, all DEMs were aggregated to a horizontal resolution of 1° × 1°, while preserving the minimum depth within the cell. This prevented the loss of oceanic islands and the generation of erroneous response curves during model calibration. Climatic variables missing data within our shallow marine masks were approximated using the mean value of all neighbouring cells (3 × 3 focal grid).

**Habitat suitability modelling**. To estimate the distribution of climatically suitable habitat for warm-water coral reefs from the Anisian to the Piacenzian, we implemented habitat suitability modelling (HSM) using MaxEnt v. 3.4.4[50,51], one of the best performing HSM methods[103]. HSM is a method in which field observations are related with environmental predictor variables based on statistically derived response curves, and is also known as habitat distribution modelling, ecological niche modelling, species distribution modelling or climatic envelope modelling[104,105]. We used present-day abiotic conditions to calibrate the HSM, constraining the tolerances of warm-water coral reefs. Subsequently, we hindcasted our HSM onto stage-level (Anisian–Piacenzian) estimates of past abiotic conditions to approximate the geographic distribution of climatically suitable habitats. This approach allowed us to constrain the tropical/subtropical climatic conditions that limit coral reefs today. For model calibration, we followed recommended practise and approximated realistic response curves[55,106]. To do so, we enabled only linear and quadratic features. In addition, we disabled model clamping, but allowed for model extrapolation when hindcasting the HDM to past environmental conditions[107]. This was necessary given that coral reefs reside at the upper thermal limit of the oceans today. HSMs were run with 100 bootstrap replications with 85% of occurrences used for model training, while 15% were reserved (randomly seeded) for model testing. Background points ($n = 10,000$) were selected from the entire study area (0–200 m substrate depth) to define available environments during model calibration. The maximum number of iterations was set to 5000 and output format changed to logistic. All other parameters were run at their default value.

The HSM was validated using two metrics: the conventional AUC statistic and the continuous Boyce index[53,54]. The AUC statistic is derived from receiver operating characteristic analyses[52]. This metric provides an overall measure of the discriminatory capacity of the model. In general, evaluation criteria for AUC values have been interpreted as good (0.9–1.0), moderate (0.7–0.9), and poor (0.5–0.7), whereas a score of <0.5 is worse than one would expect from a random model[52,108]. The continuous Boyce index provides a measure of how much model predictions differ from random distributions of observed presences across the prediction gradient[53,54]. Scores for the Boyce Index range between −1 and +1, with positive values indicating that predictions are consistent with the observed distribution of presences, whereas values of less than zero indicate that the model is worse than a random model[53,54].

In preparation of post-modelling analyses, the median suitability grid for each stage was converted to binary suitability maps (presence/absence) using two threshold approaches: 'MaxSSS' (maximising the sum of sensitivity and specificity) and 'Least Training Presence', with the former considered best practise for presence-only models[109,110]. The use of these two different binary thresholds provides the opportunity to assess the sensitivity of the results to threshold selection. In addition, Multivariate Environmental Similarity Surface analyses[55] were conducted to identify areas of novel environmental conditions across model hindcasts.

**Comparisons with empirical data**. We evaluated the capacity of model hindcasts to correctly predict fossil coral reef localities for each stage (e.g. intersecting Piacenzian fossil coral reef localities with the Piacenzian hindcast, etc.). To do so, we integrated data from the PaleoReef Database (PARED; https://www.paleo-reefs.pal.uni-erlangen.de)[111] with the Paleobiology Database (PBDB; https://www.paleobiodb.org/) to maximise the number of fossil samples available for evaluating model hindcasts. All data from PARED were downloaded and subsequently filtered to exclude reefs not identified as coral reefs. As we were principally interested in shallow marine warm-water coral reefs, we also excluded all cold-water coral reef entries, and those not identified as a 'true reef'. Subsurface reef occurrences based on seismic data were also excluded as it is difficult to determine whether or not they constitute 'true reefs'. Scleractinian coral collection data were downloaded from the PBDB with geological context set to 'reef'. Subsequently, we filtered occurrence data classified as 'perireef' or 'subreef', and data with suspect lithologies not representative of 'true reefs' (e.g. 'marl'). The PARED and the PBDB are partially linked via unique collection numbers. Using these data, we further filtered the PBDB to remove any occurrence data that we had previously excluded from the PARED. Subsequently, occurrence data from both databases were binned into stratigraphic stage-level bins and palaeorotated to their respective stage using their modern coordinates, and the Getech plate rotation model[98,102]. For temporally unconstrained data, fossil reef occurrences were assigned to a stage bin if that bin contained more than 50% of the geological time range associated with that occurrence, otherwise the data were excluded. Finally, to correspond with model hindcasts, we spatially subsampled collections and clipped data based on their palaeocoordinates at a horizontal resolution of $1° \times 1°$. The final dataset contains 535 unique fossil coral reef localities. While an overall good sample size, it should be noted that the number of reef localities varies through time, with several stages having a low sample size (Table S2–S3).

As the areal extent of estimated suitable habitat increases, the probability of accurately predicting a fossil reef locality also increases. As such, by designating large areas of the Earth as suitable in binary predictions, high predictive accuracy could be predicated. Therefore, we tested the null hypothesis that there is no difference between the predictive accuracy of fossil reef localities and those expected under a random distribution. To do so, 1000 sets of random spatial points were generated for each stage, with each set of points equal to the number of fossil coral reef localities within the respective stage. From these sets, the mean percentage of 'accurately' predicted random points (i.e. points intersecting with suitable areas) was calculated. Subsequently, one-sample (one-sided) Wilcoxon rank-sum tests were carried out to determine whether the percentage of randomly generated points intersecting with suitable habitats were significantly less than those of fossil reef localities ($\alpha = 0.05$). To allow for potential uncertainty in palaeorotations, we also afforded a search buffer around reef localities, and evaluated whether any neighbouring cells in all directions were predicted as suitable. However, all statistical comparisons were restricted to results without a search buffer. As the predictive performance of binary predictions is also subject to binary threshold selection, we were also interested in whether suitability values associated with fossil reef localities were greater than those expected under random distributions. Therefore, we tested the null hypothesis that there is no difference between the suitability values associated with fossil reef localities and those expected under a random distribution. To do so, we carried out two-sample (one-sided) Wilcoxon rank-sum tests to determine whether suitability values associated with fossil reef localities from our continuous predictions were significantly greater than those expected under a random distribution ($\alpha = 0.05$). To achieve this, we generated 1000 random spatial points for each stage within the bounds of the study area and extracted suitability values for both fossil reef sites and random points, using the median suitability grid for each stage.

**Spatiotemporal patterns**. To quantify temporal shifts in the palaeolatitudinal distribution of suitable habitat area from the Anisian–Piacenzian, the weighted centroid of suitable habitat area was determined from binary projections. To do so, the centroid of suitable cells for each stage was calculated with weights proportional to the area of each cell. This was necessary to correct for the influence of varying cell area with latitude. To prevent hemispheric differences influencing centroid calculation, we computed Northern and Southern Hemisphere centroids separately. We also calculated the palaeolatitudinal reef zone for each stage to quantify the latitudinal range of climatically suitable habitat area for coral reefs. This was quantified as the most poleward suitable cell in binary predictions for both the Northern and Southern hemispheres. This metric provides an estimation of the upper latitudinal limits of coral reef development and reef-associated biota. Finally, the areal extent of suitable habitat was calculated at global scale and within 20° palaeolatitudinal bins along the entire time series. To evaluate whether changes in the availability of climatically suitable habitat explain differences in the number of observed fossil reef sites, we implemented ordinary least-squares regression of the number of fossil coral reef sites against the global availability of climatically suitable habitat area.

## Data availability
The data generated in this study have been included within the paper, its supplementary material and the Zenodo code repository (https://doi.org/10.5281/zenodo.6458366).

Climate model simulations can be accessed at: https://www.paleo.bristol.ac.uk/ummodel/scripts/papers/.

## Code availability
All simulations and analyses were performed in R v. 4.0.3 and are available on GitHub (accessible via: https://github.com/LewisAJones/Coral_Reef_Distribution) or via the linked repository on Zenodo (https://doi.org/10.5281/zenodo.6458366)[112].

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

## Acknowledgements
We are grateful for the efforts of all those who have collected fossil coral data and have entered these data into the Paleobiology Database and the PaleoReef Database. We would also like to thank Getech PLC for supplying the digital elevation models and palaeo-rotations used in this study. L.A.J. was funded by the European Research Council under the European Union's Horizon 2020 research and innovation program (grant agreement 947921) as part of the MAPAS project. PDM's contribution was supported by a Royal Society University Research Fellowship (UF160216). A.F. and D.J.L. acknowledge NERC grant NE/K014757/1 and NE/P013805/1. This is Paleobiology Database official publication number 425.

## Author contributions
L.A.J. conceived and designed the project; L.A.J. performed the analyses; L.A.J. and P.D.M. conducted the interpretation of the data; A.F., F.B. and D.J.L. provided the GCM climate data; L.A.J., P.D.M., A.F. and D.J.L. contributed to the writing of the paper; L.A.J. produced the figures.

## Competing interests
The authors declare no competing interests.
