## [Peer Review File · Nature Communications]

REVIEWER COMMENTS

Reviewer #1 (Remarks to the Author):

This manuscript revisits the question in how far the latitudinal distribution of reefal ecosystems can provide a proxy for global paleoclimate. Jones et al. use the techniques of species distribution modeling (SDM) to provide an estimate of suitable habitat distribution for coral-dominated reefal ecosystems. The authors find that climate is a reasonable driver of reef distributions, although important exceptions exist in space and time (e.g., the earliest Jurassic and Early Cretaceous).

This is a novel approach and certainly worth publishing, when the models are better constrained. While basic software (e.g. MaxEnt) and data sources (modern reef distribution data, modern environmental data) are fine, there is a high risk that controlling parameters have not been fully considered. This may strongly affect the results and thus requires substantial revision of the manuscript.

The most important ignored variable is nutrient concentration. High nutrient levels, through coastal upwelling, rather than temperature prevent reef building along the West African and West American coasts. This problem is evident from Fig. S1 and S2. Likewise, sedimentary regimes might be crucial. Reefs will not develop under high siliciclastic shedding such as today in the Bay of Bengal etc.

It is circular reasoning to focus too much on paleoclimate drivers in the discussion, because the model was developed with paleoclimate as dominant driver. If the authors have no option to estimate nutrient levels from the paleoclimate models (although they should) and sedimentary regimes from published lithofacies maps, they could still save this paper by emphasizing more on matches and mismatches of paleoclimate and reef distributions (see regression suggestions below).

Specifically on methods:

1. The modern reef dataset appears fine and realistically show well-known tropical reef distributions. However, Fig. 1 has some issues, which are not represented in the original data set. For example, there are not reefs shown in the northern Red Sea, which is actually rich in reefs. Many reef localities along the west Indian coast are not reflecting reality and reefs along the East Australian coast appear to go too far south. Please triple check as reefs in marginal settings may strongly affect the model.
2. Regarding environmental data, I would think that siliciclastic shedding and coastal upwelling are key parameters determining the suitable habitat of reefs. Only when these parameters are included can the modern reef distribution be fully explained (e.g., the absence of coral reefs along the coast of West Africa).
3. The authors appear to re-invent the wheel in producing their own paleoclimate models. However, in the supplement they refer to Bridge system. The climate simulations there are (to my knowledge) from Paul Valdes who is however not among the authors and not acknowledged. Do those models not also predict upwelling zones? These could be used to fine-tune the reef distribution model.
4. The authors claim that fossil reef localities were taken from the Paleobiology database (PBDB). While this is ok for paleontologically described reef systems (e.g., with species identification of

reef builders), it will provide an incomplete depiction of observed reef distributions compared with the PaleoReefs database (PARED, <https://www.paleo-reefs.pal.uni-erlangen.de/>). This database has the advantage that coral reefs can be easily distinguished from other reef systems, which is less straightforward with the PBDB. And actually, in their Github code, the authors seem to be using the PARED database. All very confusing. Additionally, there might be issues with plate tectonic reconstructions. For example, high paleolatitude reefs (>50°N) in the latest Triassic are from the Pamir mountains, indicated to be at 62-63°N in the authors' occurrence table but at 31°N according to the PBDB reconstructions. So the Getech reconstructions seem to have issues.

5. The performance of the model is tested against random" points, but it is unclear what random refers to. Is this constrained by land and deep sea or not? If not, any model constraining for shallow shelf will perform better. If I were to throw randomly the actually observed number of reefs within the shallow tropical to subtropical shelf area, I bet my performance would be very close to the one observed by the authors' suitable habitat model. Again, without key constraints such as nutrient and sedimentary regimes, the authors' model is unlikely to be accurate.

Minor comments:

Why is there so much emphasis on the earliest Cretaceous, which is notorious for having a poor geological record?

The Hettangian mismatch shows again that suitable habitat is not enough to explain actual reef distributions. Evolution, including mass extinctions is key.

The discussion is long and covers several topics. It should be structured with subheadings. As latitudinal diversity gradients have not been explored, the discussion of that topic should be reduced (l. 194-205).

Rather than emphasizing the diversity-reef habitat link, the connections between suitable habitat and actual reef growth should be explored further. This could be done by reviewing the residuals of a regression analysis ($N_{\text{reefs}} \sim \text{Suitable habitat}$). When was paleoclimate a key driver and when were other factors such as evolutionary crises, species interactions (or lack thereof – see for example Teichert et al. 2020, Scientific Reports, doi 10.1038/s41598-020-73900-9) or chemical changes such as Mg/Ca ratios in the ocean?

l. 189: "might" is too weak here. It is well established that shallow-shelf area and thus suitable reef-habitat was reduced during Pangean times.

Limitations: Idealized CO₂ concentrations and uncertainties in paleoclimate models are perhaps not the major limitations to the results. It might rather be uncertainties in nutrient levels, the ecological preferences of ancient reef builders, the spatial distribution of siliciclastic shedding and mutual species interactions.

L. 264: Any particular reason to choose $R=0.75$ as a threshold?

Reference to Getech is not transparent. It seems like this is a company mining and selling open-access data from others. The Getech plate model seems to be identical to GPlates, but which version. Major discrepancies with the GPlates model used by the PBDB need to be solved.

Wolfgang Kiessling

Reviewer #2 (Remarks to the Author):

Dear authors,

I think that this manuscript is a nice and honest piece of paper, and that it will allow further scientific discussion on latitudinal gradients of diversity, fossil biases, marine diversity through time, etc. I do not have any substantial comment on it, so, for the first time, I am doing a review with no comments.

**have a nice week,
all the best**

Reviewer #3 (Remarks to the Author):

Currently no previous study links the distribution of coral reefs with abiotic conditions prior to Quaternary (~2.6 million years ago), and the authors have attempted a complex and ambitious examination of suitable habitat for corals back to the earliest known knowledge of them, ~250 million years ago, thus greatly extending current predictions. There is merit to the approach and a need to understand past global distributions, which are corroborated here using known fossil record coral reef locations giving insights into the influence of plate tectonics (shallow substrate available) and climatic conditions (temperature) that steepened the latitudinal biodiversity gradient towards the tropics.

I recommend the manuscript of Jones et al. to be considered positively for publication in Nature Climate Change, given successful revisions or explanations. My positive evaluation is based on a comprehensive assessment of both strong and weak points of the paper.

Strong points

- 1. The study proposed a method to link the distribution of coral reefs with abiotic conditions to identify suitable habitat for stony corals back to the earliest known knowledge of them, ~250 million years ago. The approach is potentially applicable over taxonomic groups and ecosystems.**
- 2. The objectives fall within the journal scope. The main results and conclusions are of great interest, appealing to the broad readership. The implications connect present-day findings with the geological past which is an important step toward understanding the drivers of macroevolutionary patterns in ancient reefs.**
- 3. The study used large-scale global data sets. Both present day coral reef locations, fossil records and paleoclimate simulations were involved in the analysis. These would have been achieved by great effort through cross-institution collaboration.**
- 4. Generally, the manuscript is well written, easily understandable and the figures presented well.**

Weak points

- 1. Lack of a good fossil record sample size in some of the stages (only 1 or 2 in some cases). This limitation may be due to sampling biases that are discussed in detail and thus I do not**

consider the paper is to be rejected only by this point. The lack of information for some stages should be made explicit when providing a single overall percentage estimate of hindcast model performance across all stages.

2. Low resolution of the climatic data (2.5x3.75 degree) and interpolated to 1x1 degree is a limitation of the study and was not considered in the discussion (1x1 degree with regards to bathymetry is mentioned only).

3. Some additional limitations of the modelling method and associated assumptions are not considered, e.g. see Couce et al 2013; Svenning et al 2011). It would have been good to consider use of a second (P/A?) model algorithm (but see below as it is not clear if MAXENT was used as presence-only framework or not from the methods as written, i.e. non-reef cells considered as absences or if used pseudo-absences) or ensemble approach (e.g. Biomod2) as congruence between models provides better confidence in the predictions, but again I do not consider that the paper is to be rejected without a second model presented, but potential differences in predictions depending on algorithms used should be considered.

The strong points support the publication merit of the paper. Some of the weak points may be overcome by revisions or justified by further explanations. Hence, I consider that the manuscript has a potential of publication at least at this stage.

Here is a list of issues, concerns, and questions, which I hope will be materials for improvement of the paper.

Introduction:

Line 32 Introduction – ocean acidification not mentioned as cause of coral reef decline but considered important factor alongside temperature induced bleaching.

Results –

LN84 AUC 0.945 – see comment below regarding use of AUC for presence-only (p-only) methods and, if used in MAXENT in p-only framework then not true AUC as originally described (see below, LN309).

LN86 – perhaps provide the predictive performance standard deviation/range here as well and note that percentages are based on very small fossil record sample sizes for some stages.

LN89 ditto

Line 89-90 – is this statement significant and does this statement relate to Figure 4 and table S2/S3?

LN 115 for someone not hugely familiar with the different stages of geological time I find it hard to relate the geological time names in the text to the Figures, i.e. Priabonian (late Eocene) here is not referred to in Figures 3 or Figures S4 and difficult to identify without looking up times in relation to the stages actually shown on the Figures – also the Figures do not define the abbreviated stages in the captions. This could be made easier for the reader to follow throughout. See also LN134 Priabonian, LN139 Serravallian (middle Miocene) etc.

LN122 'J/K boundary' relates clearly to the graphs and helps the reader follow – please consider using similar to address the above comment. Perhaps include a supplementary Figure identifying all the stages used in relation to the 5 time periods used for all the graphs (Tr, J, K, Pg, Ng) and ensure all names for historic time periods that are used in the text appear clearly in this figure.

Discussion:

LN188 repetition 'reduction in shelf area'

Include resolution of the HadCM3L climatic data as a potential limitation (see comment below in methods section)

Include limitation of low sample size for model hindcast testing for some stages (only 1 or 2 fossil records, see below LN330).

LN101 Poorer predictive performance in last 37myr, why could this be?

LN107 MESS analyses – 44 localities intersect with sea surface temperatures (SST) above present day– what are the implications for future climate change how do these relate to future predictions/scenarios – how much higher are these SST values to what we see today?

Methods:

Appropriate data modifications (occurrences and environmental data) were made for SDM modelling, e.g. data repository used, reduce collinearity and overfitting, preserving minimum depth within cells, enable quadratic and hinge features and disable clamping in MAXENT, comparison of thresholds, good modelling replication etc. Although, resolution of the HadCM3L is low (2.5x3.75degree) and interpolated to 1x1 degree – perhaps needs to be noted in the discussion as potential limitation. LN 272/275 – 278 would be better suited as a discussion point to help validate its use rather than in the methods.

LN 278 Noted that CO2 concentration held constant at different stages – what is the reasoning behind these values used and where did they come from (ref?).

LN292 Section Habitat distribution modelling. Were cells without WCMC reefs used as absences? Were pseudo-absences used instead – this was not clear, please detail in methods.

LN 309 AUC often inflated when used in presence-only models. AUC derived by MAXENT when run as a p-only framework is not AUC as it was originally defined (see Yackulic et al., 2013). Have the authors considered an alternative metric (if no true absences were used – see comment above) such as continuous Boyce Index (e.g Hirzel et al., 2006) as a complementary metric to assess p-only model performance reliably?

LN 330 Hindcast evaluation fossil records, n = 592. Table S2 indicates that the numbers per stage level are very low for some (1-2 records), which is not considered in the discussion and

limitations section of the paper when examining model performance for these time periods.

LN338 One-sample Wilcoxon signed-rank test results - was this using 'standard' or 'buffered' reef localities?

Minor - proofing

LN286 - Ensure consistency in naming deep time / deep-time

Table S1 – what does the Max Ma / Mid Ma / Min Ma relate to? Ma - one-million years?

Table S2 – LPT, avoid use of acronyms in table and figure captions

Figure captions: Abbreviated stages not defined in captions - only Tr and Ng in Figure 3. Also some figures vary in what is defined and how the stages are defined, check also between the supplementary figures to main text figures.

References:

Couce, E., Ridgwell, A. & Hendy, E. J. Future habitat suitability for coral reef ecosystems under global warming and ocean acidification. *Glob. Change Biol.* 19, 3592–3606 (2013).

Hirzel, A. H., Randin, C., and Guisan, A. 2006. Evaluating the ability of habitat suitability models to predict species presences, 9: 142–152.

Svenning, J.C., Fløjgaard, C., Marske, K.A., Nógues-Bravo, D. and Normand, S., 2011. Applications of species distribution modeling to paleobiology. *Quaternary Science Reviews*, 30(21-22), pp.2930-2947.

Yackulic, C.B., Chandler, R., Zipkin, E.F., Royle, J.A., Nichols, J.D., Campbell Grant, E.H. and Veran, S., 2013. Presence-only modelling using MAXENT: when can we trust the inferences? *Methods in Ecology and Evolution*, 4(3), pp.236-243.

Signed Joanna Bluemel

REVIEWER COMMENTS

Reviewer #1 (Remarks to the Author):

This manuscript revisits the question in how far the latitudinal distribution of reefal ecosystems can provide a proxy for global paleoclimate. Jones et al. use the techniques of species distribution modeling (SDM) to provide an estimate of suitable habitat distribution for coral-dominated reefal ecosystems. The authors find that climate is a reasonable driver of reef distributions, although important exceptions exist in space and time (e.g., the earliest Jurassic and Early Cretaceous).

This is a novel approach and certainly worth publishing, when the models are better constrained. While basic software (e.g. MaxEnt) and data sources (modern reef distribution data, modern environmental data) are fine, there is a high risk that controlling parameters have not been fully considered. This may strongly affect the results and thus requires substantial revision of the manuscript.

The most important ignored variable is nutrient concentration. High nutrient levels, through coastal upwelling, rather than temperature prevent reef building along the West African and West American coasts. This problem is evident from Fig. S1 and S2. Likewise, sedimentary regimes might be crucial. Reefs will not develop under high siliciclastic shedding such as today in the Bay of Bengal etc.

Firstly, we would like to thank the reviewer for their constructive criticism and time spent reviewing our manuscript. We agree with the reviewer that nutrient concentrations are important for constraining the distribution of coral reefs. However, in our manuscript, we are specifically testing whether a model driven by climatic factors (and the availability of shallow marine substrate) can be used to predict the distribution of fossil coral reefs. As such, the inclusion of additional variables is not particularly pertinent for our study. The aim is not to make the most constrained and well-fitted model, but to see if climatic factors can explain the distribution of coral reefs in the geological past.

While we hope that more complex models can be developed in the future to estimate the distribution of coral reefs (perhaps with more discrimination to help prospecting efforts), to our knowledge, data on nutrient concentrations at global scales for deep time are not currently available. Likewise, global-scale data on sedimentary regimes across our 250-million-year time series are also not currently available. However, we note that these variables also have little

impact on model predictions at global scale when analyses are run at a resolution of 1° x 1°, as demonstrated in a previous study on modern warm-water coral reefs (i.e. ref¹). This study also found the West African and West American coasts were favourable, even when directly including nutrient concentrations. The allocation of suitable habitats on the West African and West American coasts is also a function of the thresholds used to convert continuous predictions to binary outputs, which always represent a trade-off between sensitivity and specificity in the model. Ultimately, a model closely fitted (i.e., over-fitted) to the modern-day data will offer little transferability when projecting to different spatial or temporal settings. As such, we have not included any additional variables (e.g. nutrient concentrations). Nevertheless, we have clarified in the discussion that our results may be influenced by their omission.

It is circular reasoning to focus too much on paleoclimate drivers in the discussion, because the model was developed with paleoclimate as dominant driver. If the authors have no option to estimate nutrient levels from the paleoclimate models (although they should) and sedimentary regimes from published lithofacies maps, they could still save this paper by emphasizing more on matches and mismatches of paleoclimate and reef distributions (see regression suggestions below).

We disagree with the statement that it is circular reasoning to argue for the influence of paleoclimatic drivers. Our model was calibrated using only modern climatic data and coral reef occurrences, as clearly outlined in the original submission. We use independent fossil data to test the matches and mismatches between model hindcast predictions. We have now included a regression analysis, as suggested by the reviewer, and provide more details below where relevant.

Specifically on methods:

1. The modern reef dataset appears fine and realistically show well-known tropical reef distributions. However, Fig. 1 has some issues, which are not represented in the original data set. For example, there are not reefs shown in the norther Red Sea, which is actually rich in reefs. Many reef localities along the west Indian coast are not reflecting reality and reefs along the East Australian coast appear to go too far south. Please triple check as reefs in marginal settings may strongly affect the model.

Thank you, this has now been corrected. We have now opted to use the ReefBase dataset instead of UNEP as many of the specific issues highlighted by the reviewer are present in the UNEP database (e.g. high latitude reefs along the East Australian coast). ReefBase has the advantage of providing additional information, such as the type of reef. This has allowed us to focus our analysis on ‘true coral reefs’ and is comparable to previous work e.g. refs^{1,2}. However, we note that this has had no notable impact on our overall results and conclusions.

2. Regarding environmental data, I would think that siliciclastic shedding and coastal upwelling are key parameters determining the suitable habitat of reefs. Only when these parameters are included can the modern reef distribution be fully explained (e.g., the absence of coral reefs along the coast of West Africa).

As noted above, our study specifically tests whether climate and palaeogeography can explain the distribution of fossil coral reefs in the past. Sea surface temperature and the distribution of shallow marine habitat have been shown to be the primary drivers of coral reef distribution patterns^{1,3}. We do not disagree with the reviewer that siliciclastic shedding and coastal upwelling are important variables for constraining the distribution of coral reefs, and we have now specifically mentioned these in the manuscript discussion. However, we are specifically testing whether coral reefs track shifts in climate.

3. The authors appear to re-invent the wheel in producing their own paleoclimate models. However, in the supplement they refer to Bridge system. The climate simulations there are (to my knowledge) from Paul Valdes who is however not among the authors and not acknowledged. Do those models not also predict upwelling zones? These could be used to fine-tune the reef distribution model.

This is not accurate. These palaeoclimatic simulations (using the HadCM3L) were run by Alexander Farnsworth and Dan Lunt (co-authors on the paper), who are also based at the University of Bristol within the Bristol Research Initiative for the Dynamic Global Environment (BRIDGE). Variations and subsets of these simulations have been published extensively in recent years⁴⁻¹¹. These simulations are run using stage-level palaeogeographies provided by Getech PLC. To our knowledge, while using the same climate model, the simulations ran by Paul Valdes use the PALEOMAP palaeogeographies as boundary conditions, which are at a temporal resolution of every 5 million years. These simulations are less temporally compatible with fossil data, which are generally resolved to stage level. The HadCM3L does include estimates of vertical current velocity (an indicator of upwelling). However, as aforementioned, our analyses focused on the climatic drivers. We have now clarified this in the manuscript.

4. The authors claim that fossil reef localities were taken from the Paleobiology database (PBDB). While this is ok for paleontologically described reef systems (e.g., with species identification of reef builders), it will provide an incomplete depiction of observed reef distributions compared with the PaleoReefs database (PARED, <https://www.paleo-reefs.pal.uni-erlangen.de/>). This database has the advantage that coral reefs can be easily distinguished from other reef systems, which is less straightforward with the PBDB. And actually, in their Github code, the authors seem to be using the PARED database. All very confusing.

The data ultimately used in the original manuscript were sourced from the Paleobiology Database (PBDB). Early investigation on which database would be most suitable for our means involved looking at both datasets in detail. We opted to use the PBDB over the PaleoReefs Database (PARED) as the way in which the PBDB data is stratigraphically binned enabled the model evaluation dataset to be constructed in a straightforward manner. At time of testing, we found that the PBDB had more unique fossil reef localities (1° x 1°) than PARED, enabling a larger dataset for evaluating our model hindcasts. Reference to the PARED dataset in the GitHub code appears in earlier commits, prior to the submission of the final code. When submitting our original manuscript, code without reference to PARED was also committed. Nevertheless, we agree with the reviewer that additional important information is stored in PARED, and we have now combined both datasets. This has allowed us to filter our dataset to retain only ‘true’ scleractinian coral reefs for our analyses. However, as with other modifications to our dataset, this has had no notable impact on our results or conclusions.

Additionally, there might be issues with plate tectonic reconstructions. For example, high paleolatitude reefs (>50°N) in the latest Triassic are from the Pamir mountains, indicated to be at 62-63°N in the authors’ occurrence table but at 31°N according to the PBDB reconstructions. So the Getech reconstructions seem to have issues.

We opted to use the Getech plate rotation model as it allows our study to maintain a unified framework. The digital elevation models and palaeoclimatic models used in this study are also underpinned by the same plate tectonic model as the rotations. As such, all models are coherent spatially and temporally. Use of other models would run the risk of being spatially and temporally incompatible and could induce errors. While we agree that in some instances different plate tectonic models diverge, they are generally highly correlated in terms of their rotation direction. For example, a Spearman’s correlation test between reconstructed palaeocoordinates from Getech and PALEOMAP models (for the fossil reef localities used in this study) suggests that they are generally highly correlated (paleolatitude: $\rho = 0.871$; $P < 0.0001$; palaeolongitude: $\rho = 0.949$; $P < 0.0001$). It is also entirely subjective to say that the Getech reconstructions seem to have issues. It is entirely possible that the opposite is true, and the issues are with the in-house PBDB reconstructions (Scotese/EarthByte). Ultimately, it is difficult to suggest one is better than the other. However, Getech plate rotations are regularly used in industry, and their construction benefits from accessing proprietary data. Nevertheless, we would like to note that the stated paleolatitude by the reviewer of 31°N is based on the old Scotese rotations in the PBDB. The new

rotations using the EarthByte model provide a palaeolatitude of 53.31°N. This is accessible when downloading PBDB data. We have now also included a plot in the supplementary material demonstrating the sensitivity of palaeolatitudinal reconstruction for two rotation models (Getech & PALEOMAP) (Fig. S10).

5. The performance of the model is tested against random” points, but it is unclear what random refers to. Is this constrained by land and deep sea or not? If not, any model constraining for shallow shelf will perform better. If I were to throw randomly the actually observed number of reefs within the shallow tropical to subtropical shelf area, I bet my performance would be very close the one observed by the authors’ suitable habitat model. Again, without key constraints such as nutrient and sedimentary regimes, the authors’ model is unlikely to be accurate.

We have now clarified this in the text. Random points are generated for the entire globe. We agree with the reviewer that any model constrained to shallow marine shelf in tropical/subtropical environments would perform well. Ultimately, this includes several key environment variables which explain the distribution of modern coral reefs (e.g., shallow marine environments, warm sea surface temperatures, high light availability). We note that our actual model performance (in the modern) is evaluated using the AUC statistic and the continuous Boyce Index. The performance of model projections is evaluated by the predictive accuracy, i.e., the ability to predict the presence of fossil coral reefs. The comparison of this accuracy to random points is to demonstrate that the accuracy of model projections is not due to classifying vast proportions of the globe as suitable.

Minor comments:

Why is there so much emphasis on the earliest Cretaceous, which is notorious for having a poor geological record?

We provide emphasis on the earliest Cretaceous because it shows an interesting latitudinal trend in the distribution of climatically suitable habitat. A notable decline at temperate latitudes (30–50°N) is evident across the Jurassic/Cretaceous boundary which appears quite unique within the context of the whole Mesozoic. We would argue that the fact that the earliest Cretaceous is notorious for having a poor geological record makes this observation even more interesting, as it provides an explanation for why the record may be so poor. Nevertheless, we have reduced some emphasis on the earliest Cretaceous throughout the text at the request of the reviewer.

The Hettangian mismatch shows again that suitable habitat is not enough to explain actual reef distributions. Evolution, including mass extinctions is key.

The Hettangian mismatch is largely down to binary threshold selection. Fossil reef localities under binary threshold ‘least training presence’ accurately predict the Hettangian localities, while ‘maximising the sum of sensitivity and specificity’ does not. The latter threshold tries to reduce the number of false positives at cost to true positives. As such, this is not unexpected. However, we agree that evolution and mass extinctions are also key to controlling the distribution of coral reefs over evolutionary timescales. We have added further discussion on this in the discussion.

The discussion is long and covers several topics. It should be structured with subheadings. As latitudinal diversity gradients have not been explored, the discussion of that topic should be reduced (l. 194-205).

Thank you. We have now greatly shortened discussion on the latitudinal biodiversity gradient and added subheadings.

Rather than emphasizing the diversity-reef habitat link, the connections between suitable habitat and actual reef growth should be explored further. This could be done by reviewing the residuals of a

regression analysis ($N_{\text{reefs}} \sim \text{Suitable habitat}$). When was paleoclimate a key driver and when was it other factors such as evolutionary crises, species interactions (or lack thereof – see for example Teichert et al. 2020, Scientific Reports, doi 10.1038/s41598-020-73900-9) or chemical changes such as Mg/Ca ratios in the ocean?

We have now included a regression analysis testing the relationship between the number of reef sites and suitable habitat area. This analysis demonstrated an insignificant and weak relationship between the number of reef sites and suitable habitat area ($R^2 < 0.00$, $P > 0.05$). However, it is well documented that the fossil record is spatially and temporally uneven (e.g. refs¹²⁻¹⁴). As such, results from such regression analysis may be biased by variable sampling effort and preservation making it difficult to decipher between the influence of palaeoclimate and geological/anthropogenic biases.

l. 189: “might” is too weak here. It is well established that shallow-shelf area and thus suitable reef-habitat was reduced during Pangean times.

We agree: this has now been updated.

Limitations: Idealized CO₂ concentrations and uncertainties in paleoclimate models are perhaps not the major limitations to the results. It might rather be uncertainties in nutrient levels, the ecological preferences of ancient reef builders, the spatial distribution of siliciclastic shedding and mutual species interactions.

As previously mentioned, we agree with the reviewer that such additional variables are important for constraining the distribution of coral reefs. However, our manuscript is specifically focused on testing the role of climate and palaeogeography. As such, we do not view these as direct limitations relevant to our study. Nevertheless, we have now specifically mentioned upwelling and siliciclastic shedding zones in the limitations section of the discussion.

L. 264: Any particular reason to choose $R=0.75$ as a threshold?

Typically, 0.7–0.85 is used in the ecological niche modelling community to reduce collinearity between variables, without losing potentially ecologically important variables (e.g. see ref¹⁵). This is basically a trade-off value (see ref¹⁶ for a review). However, highly correlated, but ecologically important variables should not necessarily always be disregarded.

Reference to Getech is not transparent. It seems like this is a company mining and selling open-access data from others. The Getech plate model seems to be identical to GPlates, but which version. Major discrepancies with the GPlates model used by the PBDB need to be solved.

We have now clarified this in the text and directed towards specific references (i.e. ref¹⁷). To clarify, GPlates is the software used for running palaeogeographic reconstructions, but a plate model is still required as an input (e.g. the Getech or PALEOMAP model). The Getech plate model is one developed in-house based on the available literature, but also propriety data from industry such as oil wells. Getech also uses the GPlates software, but with their own plate model. Many of the methods used by Getech (where Paul Markwick used to work) are based on ref¹⁸. It is incorrect to say the company is mining and selling open-access data from others as they have been leading many of the efforts, and frequently provide data for various research projects without cost (e.g. refs^{4,5,7,19}). Furthermore, most of these data are not actually open-access given that nearly all publications are behind a paywall.

Wolfgang Kiessling

Reviewer #2 (Remarks to the Author):

Dear authors,

I think that this manuscript is a nice and honest piece of paper, and that it will allow further scientific discussion on latitudinal gradients of diversity, fossil biases, marine diversity through time, etc. I do not have any substantial comment on it, so, for the first time, I am doing a review with no comments.

have a nice week,

all the best

We would like to thank the reviewer for their time spent reviewing our manuscript and for their positive comments.

Reviewer #3 (Remarks to the Author):

Currently no previous study links the distribution of coral reefs with abiotic conditions prior to Quaternary (~2.6 million years ago), and the authors have attempted a complex and ambitious examination of suitable habitat for corals back to the earliest known knowledge of them, ~250 million years ago, thus greatly extending current predictions. There is merit to the approach and a need to understand past global distributions, which are corroborated here using known fossil record coral reef locations giving insights into the influence of plate tectonics (shallow substrate available) and climatic conditions (temperature) that steepened the latitudinal biodiversity gradient towards the tropics.

I recommend the manuscript of Jones et al. to be considered positively for publication in Nature Climate Change, given successful revisions or explanations. My positive evaluation is based on a comprehensive assessment of both strong and weak points of the paper.

We would like to thank the reviewer for their constructive criticism and time spent reviewing our manuscript. Below we provide point by point responses to their comments.

Strong points

The study proposed a method to link the distribution of coral reefs with abiotic conditions to identify suitable habitat for stony corals back to the earliest known knowledge of them, ~250 million years ago. The approach is potentially applicable over taxonomic groups and ecosystems.

The objectives fall within the journal scope. The main results and conclusions are of great interest, appealing to the broad readership. The implications connect present-day findings with the geological past which is an important step toward understanding the drivers of macroevolutionary patterns in ancient reefs.

The study used large-scale global data sets. Both present day coral reef locations, fossil records and paleoclimate simulations were involved in the analysis. These would have been achieved by great effort through cross-institution collaboration.

Generally, the manuscript is well written, easily understandable and the figures presented well.

Thank you, we are pleased that you found the manuscript accessible and of significant importance for the field.

Weak points

Lack of a good fossil record sample size in some of the stages (only 1 or 2 in some cases). This limitation may be due to sampling biases that are discussed in detail and thus I do not consider the paper is to be rejected only by this point. The lack of information for some stages should be made explicit when providing a single overall percentage estimate of hindcast model performance across all stages.

We agree that it is important to highlight these issues. We have now explicitly stated this in the manuscript. The reviewer is correct in stating that many of these issues are due to various sampling biases reducing the available sample sizes of some stages, which is something our approach attempts to ameliorate.

Low resolution of the climatic data (2.5x3.75 degree) and interpolated to 1x1 degree is a limitation of the study and was not considered in the discussion (1x1 degree with regards to bathymetry is mentioned only).

This has now been included in the discussion under potential caveats.

Some additional limitations of the modelling method and associated assumptions are not considered, e.g. see Couce et al 2013; Svenning et al 2011). It would have been good to consider use of a second (P/A?) model algorithm (but see below as it is not clear if MAXENT was used as presence-only framework or not from the methods as written, i.e. non-reef cells considered as absences or if used pseudo-absences) or ensemble approach (e.g. Biomod2) as congruence between models provides better confidence in the predictions, but again I do not consider that the paper is to be rejected without a second model presented, but potential differences in predictions depending on algorithms used should be considered.

We have added some additional limitations suggested by the reviewer. However, it is beyond the scope of this manuscript to document all potential limitations with applying ecological niche modelling. We elect to focus in detail on the limitations pertinent to our specific study. We have also clarified our use of the MAXENT algorithm. We have opted to retain only the MAXENT (presence-background) method. This decision was based on the following:

- 1. True presence-absence methods are most suitable for local-scale studies. This is largely due to the effort required to correctly map and identify true absences. At global scale, it can be very challenging to map such true absences. Generally pseudo-absences, or background data (as in MAXENT), are used in place of absences (as in ref¹). However, these absences are assumed based on the lack of presence data and are not the same as ‘true absences’. Therefore, we do not feel presence-absence methods are suitable for our study.**
- 2. MAXENT has been shown to be one of the highest performing ecological niche modelling algorithms ref²⁰ and is routinely used in fossil studies e.g. refs^{4,5,7,21,22}.**
- 3. Recent work has demonstrated that ensemble models are not necessarily advantageous for maximising the robustness of the model²³.**

The strong points support the publication merit of the paper. Some of the weak points may be overcome by revisions or justified by further explanations. Hence, I consider that the manuscript has a potential of publication at least at this stage.

Here is a list of issues, concerns, and questions, which I hope will be materials for improvement of the paper.

Introduction:

Line 32 Introduction – ocean acidification not mentioned as cause of coral reef decline but considered important factor alongside temperature induced bleaching.

We agree: this has now been integrated.

Results

LN84 AUC 0.945 – see comment below regarding use of AUC for presence-only (p-only) methods and, if used in MAXENT in p-only framework then not true AUC as originally described (see below, LN309).

MAXENT was used in its default presence-background framework, in which background data is randomly sampled from the whole study area. This is quite different from presence-only modelling. We agree with the perspective of the reviewer in terms of the AUC statistic not representing its original description (true presences vs. true absences). However, arguably this is no different to using MAXENT or any other algorithm in a presence/pseudo-absence framework. These are not true absences either. We note that our manuscript does not diverge from any of the current literature in our use of AUC e.g. refs^{4,5,7,21,22}. Nevertheless, we have now included an additional validation metric, as suggested by the reviewer (see below).

LN86 – perhaps provide the predictive performance standard deviation/range here as well and note that percentages are based on very small fossil record sample sizes for some stages.

This has now been integrated too.

LN89 – ditto

This has also now been integrated.

Line 89-90 – is this statement significant and does this statement relate to Figure 4 and table S2/S3?

This statement does not allude to any of our specific results. It is just a statement of fact that as you increase the amount of area considered to be suitable (and hence decrease the amount of unsuitable area) in a finite space, the chance of accurately predicting a fossil locality increases.

LN 115 for someone not hugely familiar with the different stages of geological time I find it hard to relate the geological time names in the text to the Figures, i.e. Priabonian (late Eocene) here is not referred to in Figures 3 or Figures S4 and difficult to identify without looking up times in relation to the stages actually shown on the Figures – also the Figures do not define the abbreviated stages in the captions. This could be made easier for the reader to follow throughout. See also LN134 Priabonian, LN139 Serravallian (middle Miocene) etc.

We have now updated the manuscript to include period names throughout the text when stages are first mentioned. We have also now clarified the abbreviations in the figure captions. Unfortunately, including all the stage names in the figures would not be possible, and would quickly distract from the presented results.

LN122 ‘J/K boundary’ relates clearly to the graphs and helps the reader follow – please consider using similar to address the above comment. Perhaps include a supplementary Figure identifying all the stages used in relation to the 5 time periods used for all the graphs (Tr, J, K, Pg, Ng) and ensure all names for historic time periods that are used in the text appear clearly in this figure.

The stages within each of the five time periods are the standard under the geologic timescale. Similar to the chemist’s periodic table, this information is widely available to the reader. We feel it is superfluous to include such a figure in the supplementary material. However, we have now updated the manuscript to include period names throughout the text when geological stages are mentioned. We have also now clarified the abbreviations in the figure captions.

Discussion: LN188 – repetition ‘reduction in shelf area’

We have now corrected this.

Include resolution of the HadCM3L climatic data as a potential limitation (see comment below in methods section)

This has now been integrated.

Include limitation of low sample size for model hindcast testing for some stages (only 1 or 2 fossil records, see below LN330).

This has now been integrated.

LN101 Poorer predictive performance in last 37myr, why could this be?

We have now added a section in the discussion on this.

LN107 MESS analyses – 44 localities intersect with sea surface temperatures (SST) above present day– what are the implications for future climate change how do these relate to future predictions/scenarios – how much higher are these SST values to what we see today?

We have now added a section in the discussion on this.

Methods:

Appropriate data modifications (occurrences and environmental data) were made for SDM modelling, e.g. data repository used, reduce collinearity and overfitting, preserving minimum depth within cells, enable quadratic and hinge features and disable clamping in MAXENT, comparison of thresholds, good modelling replication etc. Although, resolution of the HadCM3L is low (2.5x3.75degree) and interpolated to 1x1 degree – perhaps needs to be noted in the discussion as potential limitation.

This has now been integrated.

LN 272/275 – 278 would be better suited as a discussion point to help validate its use rather than in the methods.

We appreciate why the reviewer is suggesting this specific edit. However, we feel these sentences are better placed in the methods as it is a justification of their use.

LN 278 Noted that CO₂ concentration held constant at different stages – what is the reasoning behind these values used and where did they come from (ref?).

These values are based on ref²⁴, which compiled multi-proxy atmospheric CO₂ data from various literature sources, and are used to represent different states of the climate system (icehouse vs. greenhouse). This has now been included in the manuscript.

LN292 Section Habitat distribution modelling. Were cells without WCMC reefs used as absences? Were pseudo-absences used instead – this was not clear, please detail in methods.

MAXENT was used in its default background approach, in which it samples the background cells for calibrating the model. We have now made all of this clearer in the text.

LN 309 AUC often inflated when used in presence-only models. AUC derived by MAXENT when run as a p-only framework is not AUC as it was originally defined (see Yackulic et al., 2013). Have the authors considered an alternative metric (if no true absences were used – see comment above) such as continuous Boyce Index (e.g Hirzel et al., 2006) as a complementary metric to assess p-only model performance reliably?

MAXENT was implemented in its usual presence-background approach. However, we have now added in the continuous Boyce Index as a complementary metric at the request of the reviewer.

LN 330 Hindcast evaluation fossil records, n = 592. Table S2 indicates that the numbers per stage level

are very low for some (1-2 records), which is not considered in the discussion and limitations section of the paper when examining model performance for these time periods.

This has now been made clearer in the text.

LN338 *One-sample Wilcoxon signed-rank test results - was this using 'standard' or 'buffered' reef localities?*

We were conservative with our approach and only used the 'standard' reef localities (see Table S2 and Table S3). We have now made this clearer in the text.

Minor proofing

LN286 - *Ensure consistency in naming deep time / deep-time*

This is due to grammatical difference. The hyphen is used in 'deep-time' to serve as a single adjective before a noun, e.g. "deep-time patterns" versus "patterns in deep time".

Table S1 – what does the Max Ma / Mid Ma / Min Ma relate to? Ma - one-million years?

Ma refers to millions of years before present and is a standard abbreviation in geological/macroevolutionary literature, but it has now been defined in the table caption.

Table S2 – LPT, avoid use of acronyms in table and figure captions

This has now been revised accordingly.

Figure captions: Abbreviated stages not defined in captions - only Tr and Ng in Figure 3. Also some figures vary in what is defined and how the stages are defined, check also between the supplementary figures to main text figures.

The abbreviations of the periods have now been included.

References:

Couce, E., Ridgwell, A. & Hendy, E. J. Future habitat suitability for coral reef ecosystems under global warming and ocean acidification. Glob. Change Biol. 19, 3592–3606 (2013).

Hirzel, A. H., Randin, C., and Guisan, A. 2006. Evaluating the ability of habitat suitability models to predict species presences, 9: 142–152.

Svenning, J.C., Fløjgaard, C., Marske, K.A., Nógues-Bravo, D. and Normand, S., 2011. Applications of species distribution modeling to paleobiology. Quaternary Science Reviews, 30(21-22), pp.2930-2947.

Yackulic, C.B., Chandler, R., Zipkin, E.F., Royle, J.A., Nichols, J.D., Campbell Grant, E.H. and Veran, S., 2013. c Methods in Ecology and Evolution, 4(3), pp.236-243.

Signed Joanna Bluemel

References

1. Couce, E., Ridgwell, A. & Hendy, E. J. Environmental controls on the global distribution of shallow-water coral reefs. *Journal of Biogeography* **39**, 1508–1523 (2012).
2. Kusumoto, B. *et al.* Global distribution of coral diversity: Biodiversity knowledge gradients related to spatial resolution. *Ecological Research* **35**, 315–326 (2020).
3. Kleypas, J. A., Mcmanus, J. W. & Meñez, L. A. B. Environmental Limits to Coral Reef Development: Where Do We Draw the Line? *Am Zool* **39**, 146–159 (1999).
4. Chiarenza, A. A. *et al.* Ecological niche modelling does not support climatically-driven dinosaur diversity decline before the Cretaceous/Paleogene mass extinction. *Nature Communications* **10**, 1–14 (2019).
5. Saupe, E. E. *et al.* Climatic shifts drove major contractions in avian latitudinal distributions throughout the Cenozoic. *PNAS* **116**, 12895–12900 (2019).
6. Saupe, E. E. *et al.* Extinction intensity during Ordovician and Cenozoic glaciations explained by cooling and palaeogeography. *Nat. Geosci.* **13**, 65–70 (2020).
7. Waterson, A. M. *et al.* Modelling the climatic niche of turtles: a deep-time perspective. *Proceedings of the Royal Society B: Biological Sciences* **283**, 1–9 (2016).
8. Fenton, I. S. *et al.* The impact of Cenozoic cooling on assemblage diversity in planktonic foraminifera. *Phil. Trans. R. Soc. B* **371**, 1–12 (2016).
9. Chiarenza, A. A. *et al.* Asteroid impact, not volcanism, caused the end-Cretaceous dinosaur extinction. *PNAS* **117**, 17084–17093 (2020).
10. Farnsworth, A. *et al.* Past East Asian monsoon evolution controlled by paleogeography, not CO₂. *Sci Adv* **5**, 1–13 (2019).
11. Farnsworth, A. *et al.* Climate Sensitivity on Geological Timescales Controlled by Nonlinear Feedbacks and Ocean Circulation. *Geophysical Research Letters* **46**, 9880–9889 (2019).
12. Kiessling, W. Habitat effects and sampling bias on Phanerozoic reef distribution. *Facies* **51**, 24–32 (2005).
13. Vilhena, D. A. & Smith, A. B. Spatial Bias in the Marine Fossil Record. *PLOS ONE* **8**, 1–7 (2013).
14. Benson, R. B. J., Butler, R., Close, R. A., Saupe, E. & Rabosky, D. L. Biodiversity across space and time in the fossil record. *Current Biology* **31**, R1225–R1236 (2021).
15. Araújo, M. B. *et al.* Standards for distribution models in biodiversity assessments. *Science Advances* **5**, eaat4858 (2019).
16. Dormann, C. F. *et al.* Collinearity: a review of methods to deal with it and a simulation study evaluating their performance. *Ecography* **36**, 27–46 (2013).
17. Lunt, D. J. *et al.* Palaeogeographic controls on climate and proxy interpretation. *Clim. Past* **12**, 1181–1198 (2016).
18. Markwick, P. J. & Valdes, P. J. Palaeo-digital elevation models for use as boundary conditions in coupled ocean–atmosphere GCM experiments: a Maastrichtian (late Cretaceous) example. *Palaeogeography, Palaeoclimatology, Palaeoecology* **213**, 37–63 (2004).
19. Dunne, E. M., Farnsworth, A., Greene, S. E., Lunt, D. J. & Butler, R. J. Climatic drivers of latitudinal variation in Late Triassic tetrapod diversity. *Palaeontology* **64**, 101–117 (2020).
20. Elith, J. *et al.* Novel methods improve prediction of species’ distributions from occurrence data. *Ecography* **29**, 129–151 (2006).
21. Saupe, E. E. *et al.* Macroevolutionary consequences of profound climate change on niche evolution in marine molluscs over the past three million years. *Proceedings of the Royal Society B: Biological Sciences* **281**, 1–9 (2014).

22. Jones, L. A. *et al.* Coupling of palaeontological and neontological reef coral data improves forecasts of biodiversity responses under global climatic change. *Royal Society Open Science* **6**, 182111 (2019).
23. Hao, T., Elith, J., Lahoz-Monfort, J. J. & Guillera-Arroita, G. Testing whether ensemble modelling is advantageous for maximising predictive performance of species distribution models. *Ecography* **43**, 549–558 (2020).
24. Foster, G. L., Royer, D. L. & Lunt, D. J. Future climate forcing potentially without precedent in the last 420 million years. *Nature Communications* **8**, 1–8 (2017).

REVIEWER COMMENTS

Reviewer #1 (Remarks to the Author):

This revised manuscript has improved, but there are a few issues pending. We are still facing the problem that it is unclear what a suitable habitat really comprises. Even in the modern map there are vast areas mapped as suitable (both under LTP and MaxSSS) where no coral reefs occur (most prominently along the western margins of the Americas and Africa). As previously noted, the absence of reefs in these areas is probably due to high nutrient concentrations. A rationale for not pursuing the nutrient direction is provided in the rebuttal, but is still missing in the main text (except for a brief mention in the discussion). I argue that providing a rough estimate of coastal upwelling locations would not be overly complex given that first principles predict that those occur in low latitudes along the western margins of larger continents. Indeed older models already depict such upwelling locations for the entire Phanerozoic (e.g. Golonka, J., M. I. Ross and C. R. Scotese, 1994: Phanerozoic paleogeographic and paleoclimatic modeling maps. In: Pangea: Global Environments and Resources [Embry, A. F., B. Beauchamp and D. J. Glass (eds.)]. Canadian Society of Petroleum Geologists Memoir, 17, Calgary, 17, 1-47.) and one is left wondering why newer models cannot do so.

I agree with the authors that this paper provides a reasonable advance and further details may be directed to future work. Nevertheless, it is sad that we still don't know what the real suitable habitats were in the past. Perhaps the most interesting statement is that the concentration of reefs in the Northern Hemisphere is not just a sampling artifact but also has an environmental underpinning.

The assessment of model performance follows standard techniques. But it seems that it goes only in the direction of "what is the proportion of known reefs plotting inside the suitable habitat cells" but not the other way round "how much of the suitable habitat" is actually occupied by reefs. This is understandable given the incomplete geological record, but again should be stated to not pretend an accuracy that does not exist. I also maintain my previous concern about the null hypothesis of random points. While the authors make a good case in the rebuttal about the purpose of this approach, in the main text it still reads as if their suitability models make the difference. Again, it would be really interesting to also have a performance estimate similar to throwing darts in the tropical/subtropical shelf area.

In summary, the paper has improved and can be accepted for publication after minor revisions. However, the manuscript is a rather incremental step forward given the complex climate models and painstaking data collections underlying the analyses.

Minor comments:

l. 18. In how far is the latitudinal diversity gradient relevant here? Is this related to reefs or general? That is meant by "This"?

l. 21-22. Why to the authors think so? Corals are already establishing themselves along the coast of Japan, at the expense of kelp.

l. 28 "being" missing before "the primary constraint"?

l. 61-62. Probably true that it was never more imperative to understand the vulnerability/resilience of coral reefs to climate change, but why exactly does their long-term response matter? Different scales.

I. 101-102. Performance now given in “%” but in proportion before. Homogenize and also state if values refer to AUC or Boyce index.

I. 111, 113, 115-116, 121: Do not provide p values without underlying statistics (e.g. W). Provide exact p value. Do p-values refer to a one-sided or a two-sided test?

I. 153. $R^2=0$? Please add more digits and provide p-value. Can this relationship be shown graphically, at least in the supplement?

I. 203-225. Problematic to compare declining reef habitat with biodiversity, as reef building does not require high diversity as exemplified by low-diversity Late Miocene reefs in the Mediterranean and the lack of a correlation between diversity and reef-building capacity in the Caribbean (Johnson et al. 2008, Science, 319, doi:10.1126/science.1152197).

I. 228-232. Here an earlier, these absolute reef numbers are not informative. 71 reef sites out of how many, for example? Is it the entire 535 mentioned earlier? Then 20% many be a more informative value.

I. 249. Comma before “which”

In the caveats section of the discussion, climate variability within stages should also be mentioned. I would think that this is more important than spatial grain. Think of climate variability in the last 5 myr (the average duration of a geological stage) and what this did to the distribution of coral reefs.

I. 331. Getech is mentioned here for the first time and should be explained.

I. 384-386. The reasoning for not including subsurface reefs is unclear. Locations are nearly as precise as surface reefs. If anything, some of those subsurface “reefs” may not represent true reefs (e.g., if only note in seismic interpretations).

Reviewer #3 (Remarks to the Author):

I recommend the manuscript of Jones et al. to be considered positively for publication in Nature Climate

Change. The authors have been successful in addressing earlier review points/concerns and have provided sufficient clarification and revisions. Congratulation on a great piece of work and I look forward to seeing it published soon. All the best.

REVIEWER COMMENTS

Reviewer #1 (Remarks to the Author):

This revised manuscript has improved, but there are a few issues pending. We are still facing the problem that it is unclear what a suitable habitat really comprises. Even in the modern map there are vast areas mapped as suitable (both under LTP and MaxSSS) where no coral reefs occur (most prominently along the western margins of the Americas and Africa). As previously noted, the absence of reefs in these areas is probably due to high nutrient concentrations. A rationale for not pursuing the nutrient direction is provided in the rebuttal, but is still missing in the main text (except for a brief mention in the discussion). I argue that providing a rough estimate of coastal upwelling locations would not be overly complex given that first principles predict that those occur in low latitudes along the western margins of larger continents. Indeed older models already depict such upwelling locations for the entire Phanerozoic (e.g. Golonka, J., M. I. Ross and C. R. Scotese, 1994: Phanerozoic paleogeographic and paleoclimatic modeling maps. In: Pangea: Global Environments and Resources [Embry, A. F., B. Beauchamp and D. J. Glass (eds.)]. Canadian Society of Petroleum Geologists Memoir, 17, Calgary, 17, 1-47.) and one is left wondering why newer models cannot do so.

Firstly, we would like to thank the reviewer for their constructive comments and time spent reviewing our manuscript. While we appreciate the concerns of the reviewer regarding the lack of upwelling and nutrient variables included in the model, we would like to reiterate that we are specifically testing whether a model driven by climatic factors (and the availability of shallow marine substrate) can be used to predict the distribution of fossil coral reefs. As such, the inclusion of additional variables is not particularly pertinent for our study, but it is certainly something we would like to test in a future study. The aim of our study is not to make the most constrained and well-fitted model, but to see if climatic factors can explain the distribution of coral reefs in the geological past. We of course agree with the reviewer that variables such as nutrient concentrations from upwelling regions are important for constraining the distribution of coral reefs (among other abiotic variable, biotic interactions, and larvae dispersal limitations). However, we are simply testing climatic constraints and whether coral reefs are tracking their present-day climatic tolerances through time.

I agree with the authors that this paper provides a reasonable advance and further details may be directed to future work. Nevertheless, it is sad that we still don't know what the real suitable habitats were in the past. Perhaps the most interesting statement is that the concentration of reefs in the Northern Hemisphere is not just a sampling artifact but also has an environmental underpinning.

We appreciate the reviewer's concern and agree with the sentiment, but ultimately the paper did not set out to produce coral reef distribution maps. The aim was to produce maps of the climatically suitable habitat for coral reefs. We hope that in coming years, once carbon cycle modelling has been better integrated into general circulation models, that we can produce robust coral reef distribution maps that consider a wide range of variables (e.g. aragonite saturation) that constrain the distribution of coral reefs today.

The assessment of model performance follows standard techniques. But it seems that it goes only in the direction of “what is the proportion of known reefs plotting inside the suitable habitat cells” but not the other way round “how much of the suitable habitat” is actually occupied by reefs. This is understandable given the incomplete geological record, but again should be stated to not pretend an accuracy that does not exist. I also maintain my previous concern about the null hypothesis of random points. While the authors make a good case in the rebuttal about the purpose of this approach, in the main text it still reads as if their suitability models make the difference. Again, it would be really interesting to also have a performance estimate similar to throwing darts in the tropical/subtropical shelf area.

As the reviewer notes, when evaluating our hindcasts, we only use the presence of fossil reef sites to test our predictions. Given that we do not have absence data (and cannot reliably generate such data), and even presence data is relatively incomplete, this is the only meaningful way to evaluate our predictions. As the aim of our study was to test whether coral reefs track their present-day climatic tolerances through time, we did not set out to produce the most constrained suitability model (which would include additional variables). However, we agree that this would be a fascinating study to try next if certain model limitations/assumptions can be overcome. As such, determining how much of the estimated suitable habitat is occupied by coral reefs is not particularly relevant to our work. Furthermore, when considering competitive displacement through time (i.e. from rudists) or the incompleteness of the geological record, such a metric would not be particularly informative in the context of our study. We have added additional text to the methods section of the manuscript to highlight that our models are constraining tropical and shallow marine conditions to address the reviewer’s concerns. However, as previously highlighted in the last round of revisions, we agree with the reviewer that any model constrained to shallow marine shelf in tropical/subtropical environments would perform well. Ultimately, this includes several key environment variables which explain the distribution of modern coral reefs (e.g., shallow marine environments, warm sea surface temperatures, high light availability). However, we again note that our actual model performance (in the modern) is evaluated using the AUC statistic and the continuous Boyce Index. The performance of model hindcasts is evaluated by the predictive accuracy, i.e., the ability to predict the presence of fossil coral reefs. The comparison of this accuracy to random points is solely to demonstrate that the accuracy of model projections is not due to classifying vast proportions of the globe as suitable.

In summary, the paper has improved and can be accepted for publication after minor revisions. However, the manuscript is a rather incremental step forward given the complex climate models and painstaking data collections underlying the analyses.

Thank you for taking the time to review our manuscript and your efforts to improve the work.

Minor comments:

l. 18. In how far is the latitudinal diversity gradient relevant here? Is this related to reefs or general? That is meant by “This”?

Apologies for this not being clearer, this vagueness was due to the limits on the word count of the abstract. We have updated accordingly, and the editor can decide whether the additional content is better retained or omitted. Regarding the relevance of the latitudinal diversity gradient, coral reefs are frequently referred to as the ‘rainforests of the ocean’. This is due to providing habitats that foster incredibly high levels of biodiversity. The latitudinal diversity gradient is relevant here because species associated with coral reefs would have to track any shifts in the distribution of coral reefs, or face going extinct (or adapt).

l. 21-22. Why do the authors think so? Corals are already establishing themselves along the coast of Japan, at the expense of kelp.

The reviewer makes a fair and interesting point, and there are indeed several examples of the ‘tropicalisation’ of temperate marine communities, particularly along the coast of Japan with *Acropora* (reef-forming) corals¹. However, this segment of text was referring to coral reef ecosystems (corals and associated biodiversity) as a whole. Although there is evidence of ‘fast-growing’ corals tracking such changes, other reef-forming corals require relatively more stable conditions over longer timescales. For example, massive *Porites* have a growth rate of <3 cm per year². As such, developed reef ecosystems can take hundreds of years to form. The rate of change predicted over the next century will likely prevent the formation of developed coral reef ecosystems (and the biodiversity they accumulate) until suitable stable local conditions are established. We have now clarified in the text that we are referring to the ecosystem as a whole, not just the coral.

l. 28 “being” missing before “the primary constraint”?

Thank you, this has now been implemented.

l. 61-62. Probably true that it was never more imperative to understand the vulnerability/resilience of coral reefs to climate change, but why exactly does their long-term response matter? Different scales.

Long-term has been removed from this sentence.

l. 101-102. Performance now given in “%” but in proportion before. Homogenize and also state if values refer to AUC or Boyce index.

In this segment of text, we are referring to the raw predictive performance of the model, i.e. the percentage of points accurately predicted by the model. The specific text states this clearly, and as such we have not updated from:

“Stage-level binary hindcasts demonstrate a moderate to high predictive performance, with an average of ~60–87% (standard deviation: ~19–29%) of fossil reef localities accurately predicted by model hindcasts, depending on the stage and binary threshold selection (Fig. 2; Table S2–S3; Fig. S2–3).”

l. 111, 113, 115-116, 121: Do not provide p values without underlying statistics (e.g. W). Provide exact p value. Do p-values refer to a one-sided or a two-sided test?

In this section of the manuscript, we are summarising results for the entirety of our study period. Exact p-values and underlying statistics are provided in the supplementary material. We have now made this clearer and pointed the reader towards the supplementary tables provided. We have also clarified where we have implemented one-sided and two-sided tests.

l. 153. R2=0? Please add more digits and provide p-value. Can this relationship be shown graphically, at least in the supplement?

We have now added a graphical representation of this relationship in the supplementary at the request of the reviewer. We have also added more digits here and provided the exact p-values.

l. 203-225. Problematic to compare declining reef habitat with biodiversity, as reef building does not require high diversity as exemplified by low-diversity Late Miocene reefs in the Mediterranean and the lack of a correlation between diversity and reef-building capacity in the Caribbean (Johnson et al. 2008, Science, 319, doi:10.1126/science.1152197).

We agree with the reviewer that reef building does not require high biodiversity. However, the presence of coral reefs does promote the accumulation of biodiversity, and they support levels of productivity several hundred times higher than that of surrounding nutrient-poor areas (e.g.

ref³). Furthermore, several studies have shown a strong relationship between reef area and reef-associated biodiversity (e.g. ref⁴⁻⁶).

l. 228-232. Here an earlier, these absolute reef numbers are not informative. 71 reef sites out of how many, for example? Is it the entire 535 mentioned earlier? Then 20% many be a more informative value.

Thank you, this has now been updated to depict that it is from the whole 535 reef sites.

l. 249. Comma before “which”

The word which does not seem to appear on this line number. However, we have been through and checked all instances of the word and updated were necessary.

In the caveats section of the discussion, climate variability within stages should also be mentioned. I would think that this is more important than spatial grain. Think of climate variability in the last 5 myr (the average duration of a geological stage) and what this did to the distribution of coral reefs.

This caveat has now been added to our discussion.

l. 331. Getech is mentioned here for the first time and should be explained.

We have now included a link to Getech’s website to serve as an external reference for the reader. In addition, we have also included plots of the DEMs in the supplementary material, along with a further description.

l. 384-386. The reasoning for not including subsurface reefs is unclear. Locations are nearly as precise as surface reefs. If anything, some of those subsurface “reefs” may not represent true reefs (e.g., if only note in seismic interpretations).

This has now been updated with the reviewer’s suggestion.

Reviewer #3 (Remarks to the Author):

I recommend the manuscript of Jones et al. to be considered positively for publication in Nature Climate Change. The authors have been successful in addressing earlier review points/concerns and have provided sufficient clarification and revisions. Congratulation on a great piece of work and I look forward to seeing it published soon.

Thank you for taking the time to review our manuscript and your efforts to improve the work.

REFERENCES

1. Kumagai, N. H. *et al.* Ocean currents and herbivory drive macroalgae-to-coral community shift under climate warming. *Proceedings of the National Academy of Sciences* **115**, 8990–8995 (2018).
2. Lough, J. M. & Barnes, D. J. Environmental controls on growth of the massive coral *Porites*. *Journal of Experimental Marine Biology and Ecology* **245**, 225–243 (2000).
3. Hatcher, B. G. Coral reef primary productivity: A beggar's banquet. *Trends in Ecology & Evolution* **3**, 106–111 (1988).
4. Bellwood, D. R. Regional-Scale Assembly Rules and Biodiversity of Coral Reefs. *Science* **292**, 1532–1535 (2001).
5. Chittaro, P. M. Species-area relationships for coral reef fish assemblages of St. Croix, US Virgin Islands. *Mar. Ecol. Prog. Ser.* **233**, 253–261 (2002).
6. Huntington, B. E. & Lirman, D. Species-area relationships in coral communities: evaluating mechanisms for a commonly observed pattern. *Coral Reefs* **31**, 929–938 (2012).